# RNA-DNA strand exchange by the *Drosophila* Polycomb complex PRC2

Célia Alecki[1,2], Victoria Chiwara[1], Lionel A. Sanz[3], Daniel Grau[4], Osvaldo Arias Pérez [1,5], Elodie L. Boulier[1], Karim-Jean Armache[4], Frédéric Chédin [3] & Nicole J. Francis[1,2,6 ✉]

Polycomb Group (PcG) proteins form memory of transient transcriptional repression that is necessary for development. In *Drosophila*, DNA elements termed Polycomb Response Elements (PREs) recruit PcG proteins. How PcG activities are targeted to PREs to maintain repressed states only in appropriate developmental contexts has been difficult to elucidate. PcG complexes modify chromatin, but also interact with both RNA and DNA, and RNA is implicated in PcG targeting and function. Here we show that R-loops form at many PREs in *Drosophila* embryos, and correlate with repressive states. In vitro, both PRC1 and PRC2 can recognize R-loops and open DNA bubbles. Unexpectedly, we find that PRC2 drives formation of RNA-DNA hybrids, the key component of R-loops, from RNA and dsDNA. Our results identify R-loop formation as a feature of *Drosophila* PREs that can be recognized by PcG complexes, and RNA-DNA strand exchange as a PRC2 activity that could contribute to R-loop formation.

[1] Institut de recherches cliniques de Montréal, 110 Avenue des Pins Ouest, Montréal, QC H2W 1R7, Canada. [2] Département de biochimie et médecine moléculaire Université de Montréal, 2900 Boulevard Edouard-Montpetit, Montréal, QC H3T 1J4, Canada. [3] Department of Molecular and Cellular Biology and Genome Center, 1 Shields Avenue, University of California, Davis, Davis, CA 95616, USA. [4] Skirball Institute of Biomolecular Medicine, Department of Biochemistry and Molecular Pharmacology, New York University School of Medicine, New York, NY 10016, USA. [5] Natural Sciences and Engineering Postgraduate, Universidad Autonoma Metropolitana, Cuajimalpa, Mexico City, Mexico. [6] Division of Experimental Medicine, McGill University, 1001 Decarie Boulevard, Montreal, QC H4A 3J1, Canada. ✉email: nicole.francis@ircm.qc.ca

During *Drosophila* embryogenesis, transiently expressed transcription factors activate homeotic (*Hox*) genes in certain regions of the embryo and repress them in others to dictate the future body plan[1]. Polycomb Group (PcG) proteins form a memory of these early cues by maintaining patterns of *Hox* gene repression for the rest of development[1–3]. This paradigm for transcriptional memory is believed to be used by the PcG at many genes in *Drosophila*, and to underlie the conserved and essential functions of PcG proteins in cell differentiation and development from plants to mammals[4,5]. Polycomb response elements (PREs) are DNA elements that can recruit PcG proteins, but they also recapitulate the memory function of the PcG—when combined with early acting, region-specific enhancers in transgenes, they maintain transgene repression in a PcG-dependent manner only in regions where the early enhancer was not active[2,6,7]. PREs contain a high density of binding sites for transcription factors that can recruit PcG proteins through physical interactions[7]. However, the widespread expression, binding pattern, and properties of factors that bind PREs cannot explain how PREs can exist in alternate, transcription-history dependent states to maintain restricted patterns of gene expression, or how they can switch between states[2]. Furthermore, DNA sequences with PRE-like properties have been difficult to identify in other species[7–9] despite the conservation of PcG complexes, their biochemical activities, and their critical roles in development.

RNAs may provide context specificity to PcG protein recruitment and function. Some PREs, and some PcG-binding sites in mammalian and plant cells, are transcribed into ncRNA, while others reside in gene bodies, and thus are transcribed when the gene is expressed[10,11]. Both the direction and level of transcription have been correlated with the functional state of PREs[10–12]. The PcG complex Polycomb Repressive Complex 2 (PRC2) has a well-described high affinity for RNA[13–17]. RNA is suggested to recruit PRC2 to specific chromatin sites[13], but RNA binding can also compete for chromatin binding and inhibit PRC2 activity[11,17–20]. One way for RNA to interact with the genome is by the formation of R-loops, three-stranded nucleic acid structures formed when an RNA hybridizes to a complementary DNA strand, thereby displacing the second DNA strand[21]. R-loops have been linked to regulation of transcription and chromatin previously, through a variety of mechanisms (reviewed in refs. [22,23]). This includes links to PcG regulation in mammalian cells. The formation of R-loops over genes with low to moderate expression is associated with increased PcG binding and H3K27 trimethylation (H3K27me3) in human cells[24] and R-loops have recently been implicated in promoting PRC1 and PRC2 recruitment in mammalian cells[25], although other evidence suggests they antagonize recruitment of PRC2[26]. We hypothesized that R-loop formation could biochemically link RNA to PcG-mediated silencing through PREs and tested this idea in the *Drosophila* system.

Here, we identify R-loop forming sequencing in *Drosophila* embryos and S2 cells and observe that ~25% of PREs form R-loops. Interestingly, PREs that form R-loops are more likely to be bound by PcG proteins compared with PREs that do not form R-loops, suggesting that R-loops may be involved in PcG targeting. In vitro, PRC1 and PRC2 recognize R-loops and open DNA-bubbles. Further, when provided dsDNA and RNA, PRC2 induces the formation of RNA–DNA hybrids, the key components of R-loops. These data suggest a mechanism for RNA to contribute to targeting of PcG proteins via R-loop formation induced by the RNA-DNA strand exchange activity of PRC2.

## Results

### R-loops form at many PREs in *Drosophila* embryos and cells.
To determine whether R-loops form at PREs, we carried out two biological replicates of strand-specific DNA-RNA Immunoprecipitation followed by next generation sequencing (DRIP-seq) in *Drosophila* embryos (2–6 and 10–14 hour (H)) and in S2 cells (Fig. 1, Supplementary Fig. 1). DRIP-seq peaks called relative to both input and RNase H-treated control samples and present in both replicates were analyzed. Ten-positive sites were validated by DRIP-qPCR (Supplementary Fig. 1b). Nearly 3/4 of R-loops formed over annotated genes (Supplementary Fig. 1). R-loops were observed over genes encompassing all levels of transcription, although a majority were associated with genes with no or low levels of expression (Supplementary Fig. 2a, b). Most R-loops formed with the strandedness expected from annotated transcripts (Fig. 1 a, c, Supplementary Fig. 2c), as observed in other species[24,27,28].

We detected R-loops at 22–33% of PREs (Fig. 1a–c, Supplementary Fig. 1d, 2a–d, 3a–c). R-loops at PREs in embryos were more likely to form in an antisense orientation to annotated transcripts than total R-loops (Supplementary Fig. 2c). PREs that form R-loops were also more likely to overlap with RNA polymerase II than PREs that do not form R-loops (Supplementary Fig. 2d).

To test whether R-loops are related to the functional state of PREs, we used publicly available ChIP-seq datasets to compare PcG protein binding over PREs as a function of R-loop formation in each of our three samples. For each PcG protein tested, the median read density for PcG proteins was higher over PREs with R-loops than that over PREs without R-loops (Fig. 1d–f, Supplementary Fig. 2f–h). The binding of Dsp1 and GAF, two proteins implicated in PcG recruitment and in both repressive and active states of PREs[29–31], was also higher at R-loop-positive PREs. Although binding of PcG proteins to PREs is necessary for their repressive function, it may not be sufficient, since analyses of PcG protein binding at a small number of PREs in the ON and OFF states did not detect differences in PRC1 or PRC2 binding[32,33]. Instead, histone modifications at and around PREs are correlated with the functional state so that PREs in the OFF state are marked with H3K27me3[32]. In both developing embryos and S2 cells, H3K27me3 density was higher at PREs with R-loops than at those without R-loops (Fig. 1e). H3K27Ac, a mark of the active state, was found at a small number of PREs, but correlated weakly with the presence of R-loops (Supplementary Fig. 2e, i–k, 3d–f). A small fraction of R-loops that are present at PREs in early stage embryos are absent in the later stage (Fig. 1c), suggesting that some R-loops at PREs are developmentally regulated. Developmental dynamics of R-loops at PREs are likely underestimated in our experiments because we used whole embryos. To test whether transient presence of an R-loop at a PRE predicts the repressed state, we identified PREs that form R-loops in early (2–6H) but not late (10–14H) embryos and interrogated the levels of H3K27me3 in later embryonic stages (12–16H). PREs that formed R-loops in early embryos had a higher density of H3K27me3 at subsequent developmental stages than PREs that did not form R-loops at either stage (Fig. 1f); these PREs were not enriched for H3K27Ac ($p = 0.0885$).

### PRC1 and PRC2 bind R-loops and open DNA bubbles in vitro.
To understand biochemically how R-loops could promote the repressive state of PREs, we turned to in vitro assays. Recruitment of PRC1 and PRC2 to some sites in mammalian cells has recently been linked to the presence of R-loops[25] so that we wondered if either complex might recognize R-loops. To measure the relative affinities of PRC1 and PRC2 for different nucleic acid substrates, we prepared dsDNA, R-loop or open DNA bubble templates from synthetic oligonucleotides corresponding to a sequence in the *vestigial* (*vg*) PRE (Supplementary Table 1), PRC2, and PRC1 (Supplementary Fig. 4a, b), and used them in EMSA experiments.

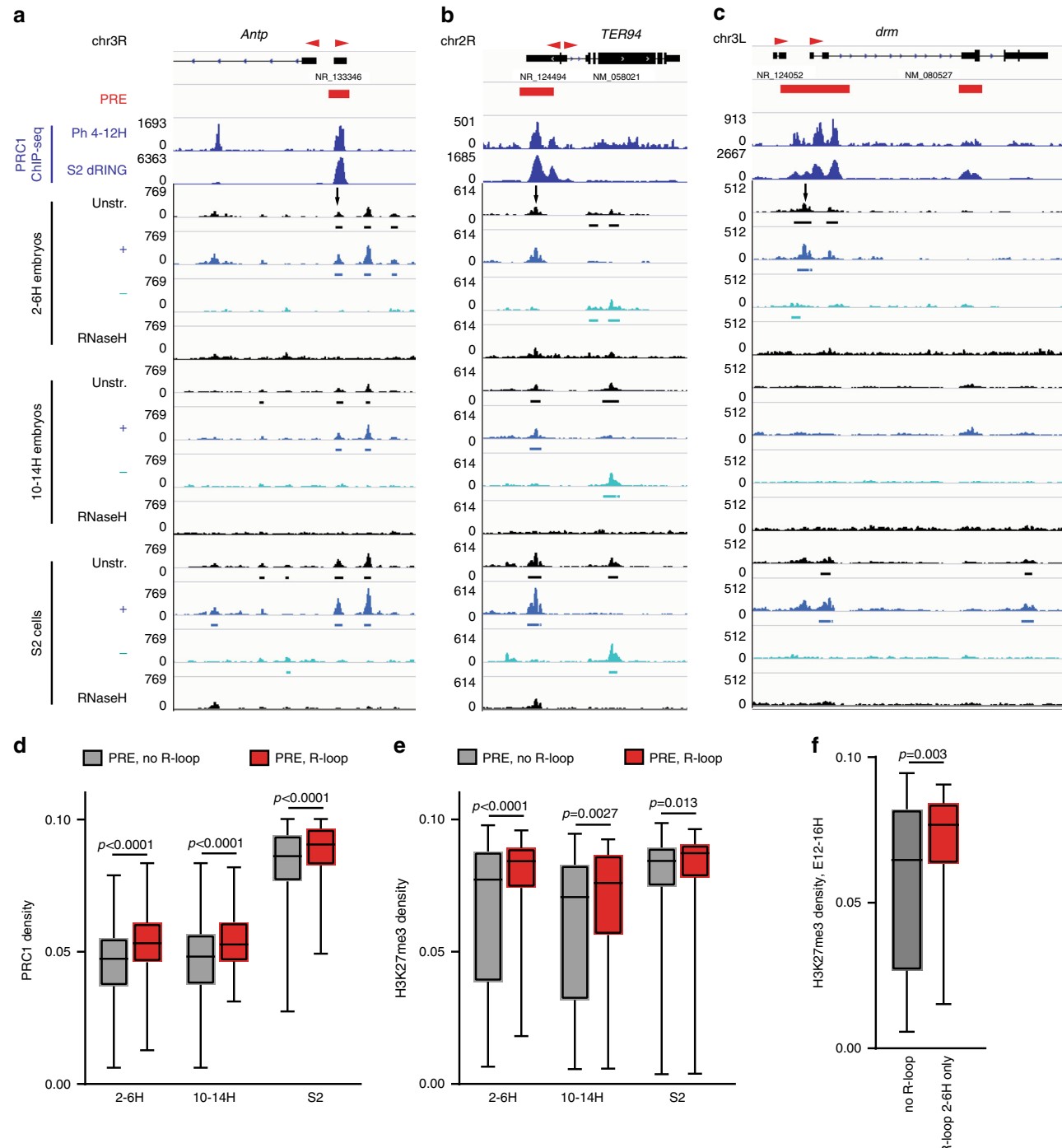

**Fig. 1 R-loops form at *Drosophila* PREs and correlate with a repressed state. a–c** DRIP-seq traces showing R-loop formation at PREs bound by PRC1 components (arrows) in 2–6H and 10–14H *Drosophila* embryos (Ph), and in S2 cells (dRING). RNaseH-treated samples are negative controls. "Unstr" indicates all R-loops, while + and – indicate strand specific tracks; direction refers to the DNA in the RNA–DNA hybrid. Called peaks are indicated under the traces. Red arrowheads above genes indicate direction of annotated transcripts. **d, e** Median normalized intensity of PRC1 components (**d**), or H3K27me3 (**e**) over PREs with or without R-loops. 2–6H and 10–14H R-loop data are compared with Ph at 4–12H and H3K27me3 at 4–8H and 12–16H respectively. S2 cell R-loop data are compared with dRING. Whiskers show min. to max. **f** Median normalized intensity of H3K27me3 at 12–16H over PREs where R-loop formation is detected in 2–6H but not in 10–14H compared to PREs with no R-loops detected at either stage. *p*-values are for two-tailed Mann–Whitney tests. See also Supplementary Fig. 2.

PRC1 lacking the Ph subunit (PRC1ΔPh) was used for these assays because this complex can be isolated in larger amounts. Our previous work indicates that PRC1 with and without Ph behave similarly in DNA and chromatin binding experiments, where binding is largely dependent on the C-terminal region of PSC[34]. EMSA experiments showed that PRC2 and PRC1 bind

more tightly to R-loops or open DNA bubble templates than to dsDNA (Fig. 2a, b). The fact that binding of PcG complexes produces a "well-shift" rather than discrete bands makes quantification of these experiments imprecise. We therefore used filter binding with the same templates to measure binding (Fig. 2c, d, Supplementary Fig. 5a, b). Under our conditions, both PRC1ΔPh

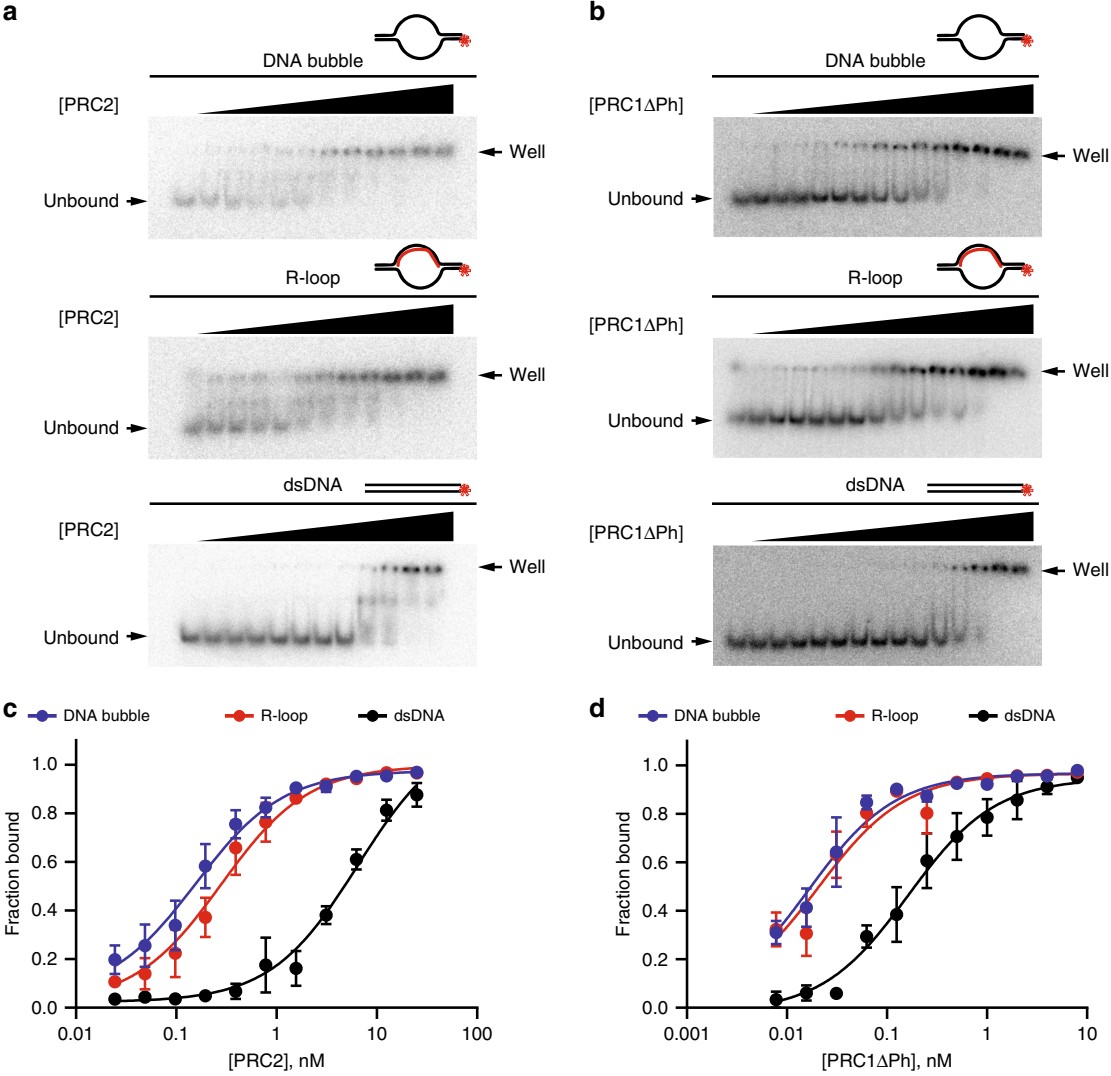

**Fig. 2 PRC1 and PRC2 recognize R-loops. a**, **b** PRC2 (**a**) and PRC1ΔPh (**b**) binding to open DNA bubble, R-loop and dsDNA oligonucleotides measured by EMSA. [PRC2] = 0.02–25 nM for DNA bubble, R-loop and dsDNA and [PRC1ΔPh] = 0.02–16 nM (active concentration). **c** PRC2 binding to DNA bubble, R-loop and dsDNA oligonucleotides measured by filter binding assay. [PRC2] = 0.02–25 nM, $n := 3$. Points show the mean +/− S.E.M. Apparent Kd(95% CI): DNA bubble = 0.16 nM (0.11–0.23); R-loop = 0.28 nM (0.20–0.41); dsDNA = 6.44 (4.26–9.99). **d** PRC1ΔPh binding to DNA bubble, R-loop and dsDNA oligonucleotides measured by filter binding assay. [PRC1ΔPh] = 0.08–8 nM, $n = 3$. Points show the mean +/− S.E.M. Apparent Kd(95% CI): DNA bubble = 0.016 nM (0.012–0.022); R-loop = 0.021 nM (0.014–0.031); dsDNA = 0.164 (0.106–0.254). Source data are provided in Source Data file.

and PRC2 bind more strongly (~8x and ~23×, respectively) to an R-loop or an open DNA bubble (~10× and ~40×, respectively) than to dsDNA. For PRC1ΔPh the Kd measured with R-loop and open DNA bubble substrates are close to the probe concentration. We were unable to lower the probe concentration due to limitations on the sensitivity of detection. Therefore, these Kds should be regarded as upper limits. Because of this limitation, the difference between the R-loop and open DNA bubble could be larger than what we measure. We conclude that PRC2 and PRC1 recognize R-loops, as well as open DNA bubbles. This suggests that PcG complexes recognize structured DNA or ssDNA rather than the RNA–DNA hybrid part of R-loops.

**PRC2 induces RNA–DNA strand exchange**. Because interactions between PRC2 and RNA are widely implicated in its regulation and function, we wondered if PRC2 might influence R-loop formation. We titrated PRC2 into reactions with radio- or fluorescently labelled RNA and the corresponding linear dsDNA (Fig. 3a–c). We observed a PRC2 dose-dependent appearance of

an RNA species that migrates at the position of dsDNA (Fig. 3d, e, g, Supplementary Fig. 6a, b). These putative RNA–DNA hybrids formed with either the sense or anti-sense RNA, but not with a non-complementary RNA, indicating that base pairing between RNA and DNA is required (Fig. 3d–g).

To confirm that the PRC2 reaction products indeed contain RNA–DNA hybrids, we tested their nuclease sensitivity: the RNA band that migrates at the position of dsDNA was fully degraded by RNase H and resistant to RNase A (Fig. 3h, i, Supplementary Fig. 6b). We also tested if the S9.6 antibody can recognize the PRC2 reaction products, as expected if they contain RNA–DNA hybrids. RNA–DNA strand exchange assays were carried out with or without RNA, the purified products were incubated with S9.6 antibody-coupled magnetic beads, and the isolated DNA analyzed on agarose gels. Nucleic acids were efficiently immunoprecipitated by the S9.6 antibody only when RNA was included in the reaction (Supplementary Fig. 6c, d). We conclude that PRC2 mediates RNA–DNA strand exchange when incubated with RNA and dsDNA.

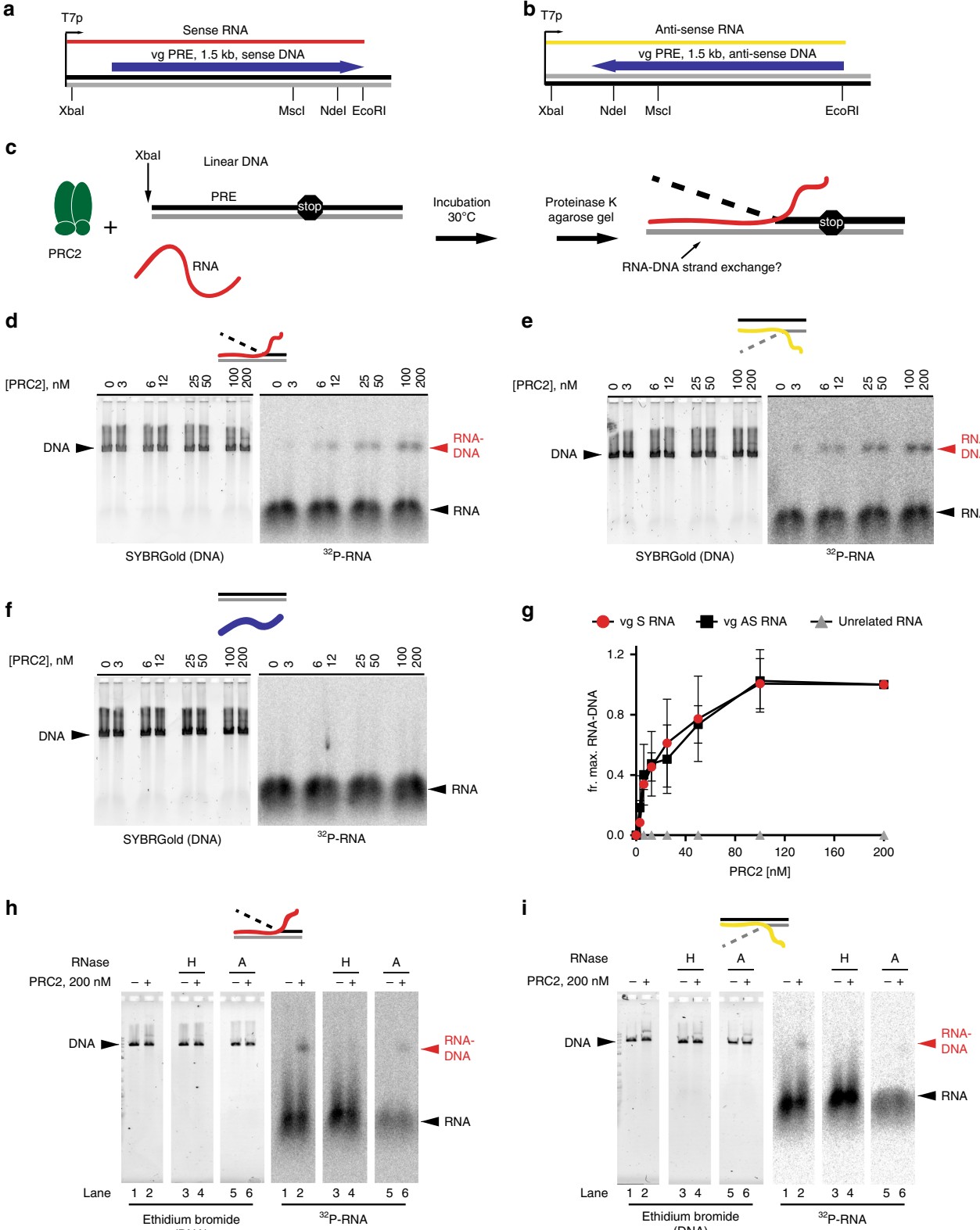

**Fig. 3 PRC2 has RNA–DNA strand exchange activity. a**, **b** Nucleic acids used for RNA strand exchange. Plasmids with the *vestigial* (*vg*) PRE in either orientation are transcribed to produce sense (**a**) or anti-sense RNAs (**b**). Linearized plasmid is used as the DNA template. "Stop" indicates where the RNA ends relative to the DNA. **c** Strand exchange assay scheme. **d**–**f** Titration of PRC2 with either the sense (**d**) or anti-sense (**e**) *vg* PRE RNA, or a non-complementary RNA (**f**). **g** Quantification of three titrations; graphs show mean +/− S.D. of the the fraction of signal with 200 nM PRC2. **h**, **i** RNA strand invasion products are sensitive to RNaseH (lane 4) but resistant to RNaseA (lane 6). See also Supplementary Fig. 6. Source data are provided in Source Data file.

Using fluorescently labelled RNAs, we estimated the extent of RNA–DNA hybrid formation. By the end of a 60-min reaction containing 3 fmol of linear DNA and 1.9 fmol of RNA, close to 40% of the DNA had undergone strand exchange with the RNA (Supplementary Fig. 7a, b). RNA–DNA strand exchange induced by PRC2 required $MgCl_2$ but not nucleotide addition (Supplementary Fig. 7c, d). To confirm that the ATP-independence of the reaction did not reflect contaminating ATP in the RNA or PRC2 preparations, we treated each with apyrase prior to carrying out RNA–DNA strand exchange assays and the results were unchanged (Supplementary Fig. 7e–g).

To confirm that RNA–DNA hybrid forming activity is specific to PRC2, we tested two control proteins, the transcription factor NFY and the PcG protein Sxc. Neither of these proteins induced formation of RNA–DNA hybrids, although they bind both DNA and RNA (Supplementary Fig. 8a–e). RNA–DNA hybrid formation activity also co-fractionated with PRC2 through size exclusion chromatography (Supplementary Fig. 8f–h).

Cellular nucleases are common contaminants when purifying chromatin-associated enzymes; the presence of nuclease contaminants in preparations of PRC2 could promote RNA–DNA hybrid formation by exposing single-stranded DNA and enabling spontaneous annealing with complementary RNA. To address this possibility, we performed three experiments. First, we incubated phosphorylated ds and ssDNA oligonucleotides with PRC2, or a series of commercially available endo- and exonucleases. Oligonucleotides were then analyzed on denaturing acrylamide gels, which were stained with SYBRGold to visualize degradation products. While oligonucleotides were fully degraded across the nuclease titrations, we did not detect degradation products after incubation with PRC2 (Supplementary Fig. 9a–f). We also tested whether exonuclease treatment can lead to RNA–DNA hybrid formation under experimental conditions used for PRC2 (Supplementary Fig. 9g–j). Exonuclease III treatment led to RNA–DNA hybrid formation, but this required enzyme concentrations that clearly degrade ds and ssDNA oligonucleotides (Supplementary Fig. 10f, j).

Second, we reasoned that if our PRC2 preparations contained nuclease activity, treatment of dsDNA with PRC2 should expose long stretches of ssDNA that could form filaments with single strand DNA-binding protein (SSB), which would be visible by electron microscopy (EM) (Fig. 4a–c). We incubated linear DNA with PRC2 or exonuclease III using the same experimental conditions leading to RNA–DNA hybrid formation, purified the DNA, incubated it with SSB, and visualized the samples by negative stain EM. SSB-coated ssDNA filaments were clearly visible in DNA samples pre-treated with exonuclease III (Fig. 4b) but not in DNA samples treated with PRC2 (Fig. 4c).

Finally, to functionally test whether PRC2 nuclease contaminants in PRC2 could account for RNA–DNA hybrid formation, we pre-treated DNA templates with PRC2, and used them in RNA–DNA hybrid forming assays in the presence or absence of PRC2 (Fig. 4d). If nuclease activity in PRC2 preparations exposes ssDNA that allows formation of RNA–DNA hybrids, the pre-treated templates should form RNA–DNA hybrids without further requirement for PRC2. However, we detected RNA–DNA hybrids only when PRC2 was added during the RNA–DNA strand exchange reaction, and not in samples in which the DNA was pre-treated with PRC2 (Fig. 4e, f). We conclude that contaminating nuclease activity in PRC2 preparations cannot explain PRC2 induced RNA–DNA hybrid formation and therefore that formation of these RNA–DNA hybrids was directly catalyzed by PRC2.

**Substrate requirements for RNA–DNA strand exchange.** To determine the DNA and RNA substrate requirements for

RNA–DNA strand exchange, we tested DNA templates with different ends (4 or 2 base pair (bp) 5′ or 4 bp 3′ overhangs, or blunt ends), prepared by digestion with different restriction enzymes. In all cases, the RNAs used overlap the end of the DNA (Fig. 5a–c). Similar levels of RNA–DNA strand exchange were observed with all types of DNA ends (Supplementary Fig. 10a, b, e). However, templates for which the RNA was internal to the DNA did not lead to hybrid formation (Fig. 5a–c and Supplementary Fig. 10c, d).

The above experiments suggested that the overlap of the RNA with the ends of the dsDNA is important for PRC2-mediated RNA–DNA strand exchange. To further analyze this, we digested DNA templates with EcoRI such that the resulting products have eight non-complementary bases prior to the start of the RNA on the bottom DNA strand, and four on the top strand (Fig. 5d–j). For each of three pairs of sense-antisense RNA–DNA combinations, RNA–DNA strand exchange is only observed when the RNA is complementary to the top strand (i.e. with four unmatched bases rather than eight). Together, these experiments suggest near complete overlap between the RNA and DNA ends is important for PRC2-mediated RNA–DNA strand exchange, implying that the reaction initiates at the DNA end, but that a specific DNA end structure is not required.

PRC2-mediated RNA–DNA strand exchange could require binding to DNA, to RNA, or to both. Detailed analyses of PRC2 binding to nucleic acids and chromatin are consistent with PRC2 making multiple contacts with both substrates[19,35], while functional assays are consistent with a single binding site that can bind chromatin, DNA or RNA, but has highest affinity for RNA so that RNA can compete for binding to DNA or chromatin[11,17,18]. To understand the role of RNA and DNA interactions in PRC2-mediated RNA–DNA strand exchange, we titrated each substrate and changed the order of addition in the reaction. Addition of RNA prior to DNA inhibited the reaction at low concentrations of PRC2. RNA–DNA strand exchange increased with increasing RNA concentration and decreased with increasing DNA concentration (Fig. 6a–f, Supplementary Fig. 11a–c).

While the exact mechanism by which PRC2 induces RNA–DNA strand exchange remains unknown, one step must be the annealing of RNA and DNA. We tested whether PRC2 enhances annealing of RNA and ssDNA and find that PRC2 can induce RNA–DNA hybrids by annealing of single-stranded oligonucleotides. PRC2 can also anneal ssDNA to form dsDNA (Supplementary Fig. 12). Unlike RNA–DNA strand exchange, RNA–DNA oligo annealing does not require $MgCl_2$.

## Discussion

The demonstration that PRC2 induces the formation of RNA–DNA hybrids in vitro, that PRC2 and PRC1 recognize R-loops in vitro, and that R-loops are present at PREs in vivo suggest a mechanistic model for how RNAs could induce or maintain the OFF state of PREs (Fig. 7). If PREs (or the gene they control and in many cases are embedded in) are highly transcribed, the RNA could compete for PRC2 binding to chromatin, as has been demonstrated in vitro and in vivo[17,18,20] (Fig. 7d). However, a lower level of transcription through a PRE (or transcription in an orientation that is favourable for R-loop formation) could allow R-loops to form, possibly via the RNA–DNA hybrid forming activity of PRC2 (Fig. 7a, c). R-loop formation will repress additional RNA production by preventing RNA polymerase passage[23] allowing recruitment of additional PRC2 (by PRE-binding transcription factors or interactions with other PcG proteins[36,37]) and its retention on chromatin. PRC2 could then modify histones to maintain a repressive chromatin state (Fig. 7c). The R-loop, in conjunction with H3K27me3 and PRE-

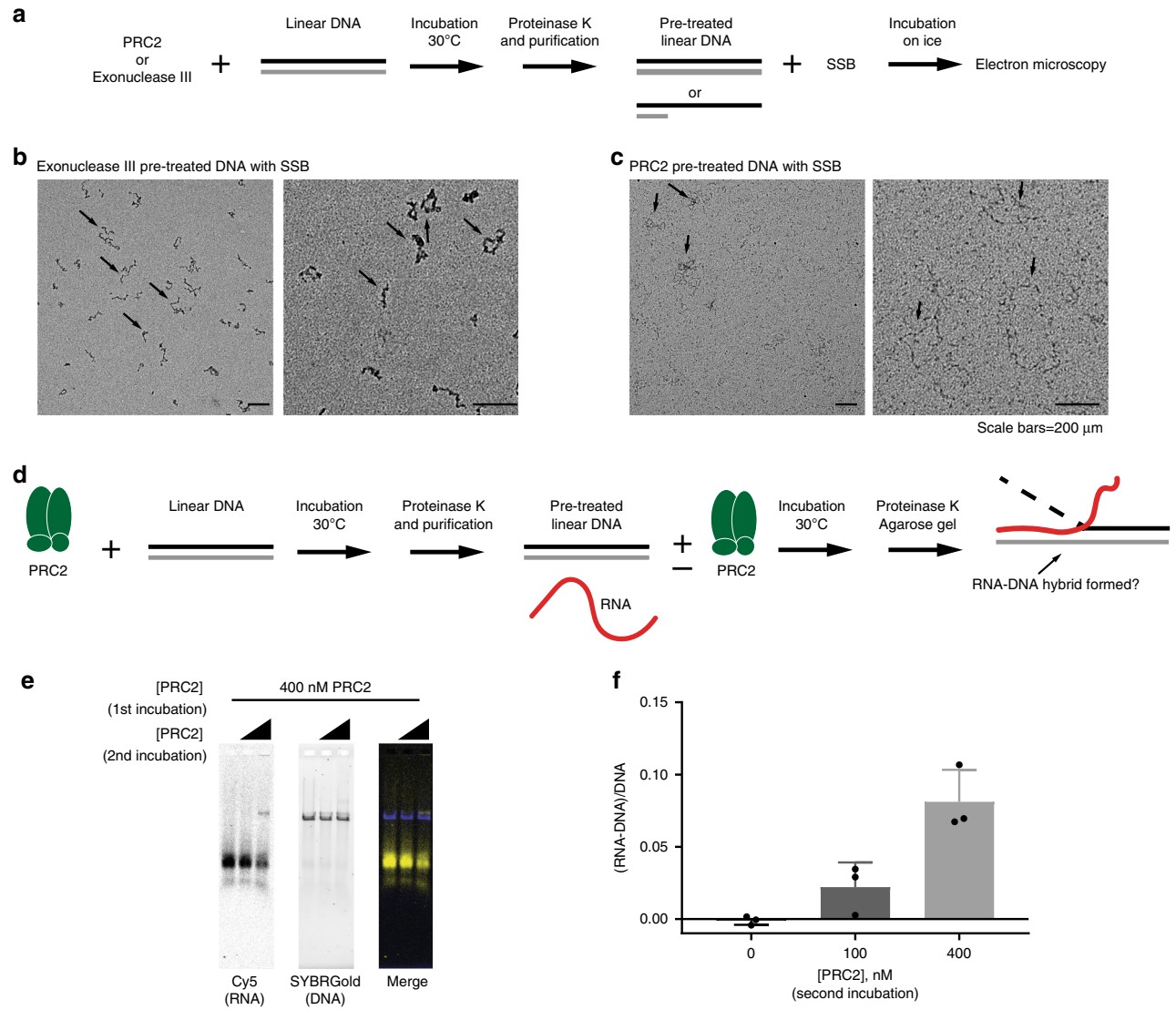

**Fig. 4 Nuclease contamination cannot explain RNA–DNA hybrid formation by PRC2. a** Scheme of experiment to test whether PRC2 generate a long ssDNA filament. Linear DNA was incubated with PRC2 or exonuclease III and purified. SSB protein was added to pre-treated DNA and samples were visualized by electron microscopy. **b, c** Representative EM pictures of DNA pre-treated with Exonuclease III (**b**) or with PRC2 (**c**). Arrows indicate DNA with (**b**) or without (**c**) SSB coating. **d** Scheme of experiment to test whether pre-incubation of DNA with PRC2 allows RNA–DNA hybrid formation in the absence of PRC2.is sufficient. **e, f** Representative gel (**e**) and quantification (**f**) of strand exchange reactions using DNA that was pre-treated with PRC2 (400 nM) as the substrate. $n = 3$. Graph shows the mean $+/-$ S.E.M. Source data are provided in Source Data file.

binding transcription factors, would also promote binding of PRC2 and PRC1 (Fig. 7b, c). R-loops may also interfere with binding or function of proteins that promote the active state of PREs, although this remains to be tested. Our data indicate that both coding and ncRNAs form R-loops. The regulation of these RNAs and therefore of R-loops could provide transcriptional memory and developmental context specificity to PcG recruitment by transcription factors that constitutively recognize PREs. A conceptually similar model for how high levels of RNA production at PREs could promote the ON state and low levels the OFF state was proposed previously[12]; R-loop formation provides one mechanism by which it can occur. Although this model is highly speculative at this time, it integrates many observations, and provides testable hypotheses.

Observations in *Drosophila* are also consistent with a possible connection between R-loops and PcG function. The helicase Rm62 interacts genetically with both PcG and TrxG genes, and colocalizes with the PRE-binding protein Dsp1 on polytene chromosomes[38]. Rm62 is the *Drosophila* homologue of the DDX5 helicase, which can unwind RNA–DNA hybrids in vitro and is implicated in R-loop resolution in vivo[39]. A recent genome-wide RNAi screen for TrxG interacting genes (which should antagonize PcG function) identified the gene for RNaseH1[40]. RNA has been suggested to be important in switching PREs between OFF and ON states[11,12,41,42], although this has been contested by experiments aiming to test whether transcription through a PRE can switch it to the active state[43,44]. Resolution of R-loops by cellular RNases or RNA–DNA helicases could contribute to switching PRE states, which will be intriguing to test. It is also likely that even in the simple model suggested in Fig. 7, the levels of RNA corresponding to "low" and "high", and the strength of the effect will depend both on the genomic context and the sequences of the RNAs that are produced.

R-loop formation is observed at ~30% of PREs; these may represent a specific class of PREs. Most R-loops are believed to form co-transcriptionally, so that R-loops would be predicted to

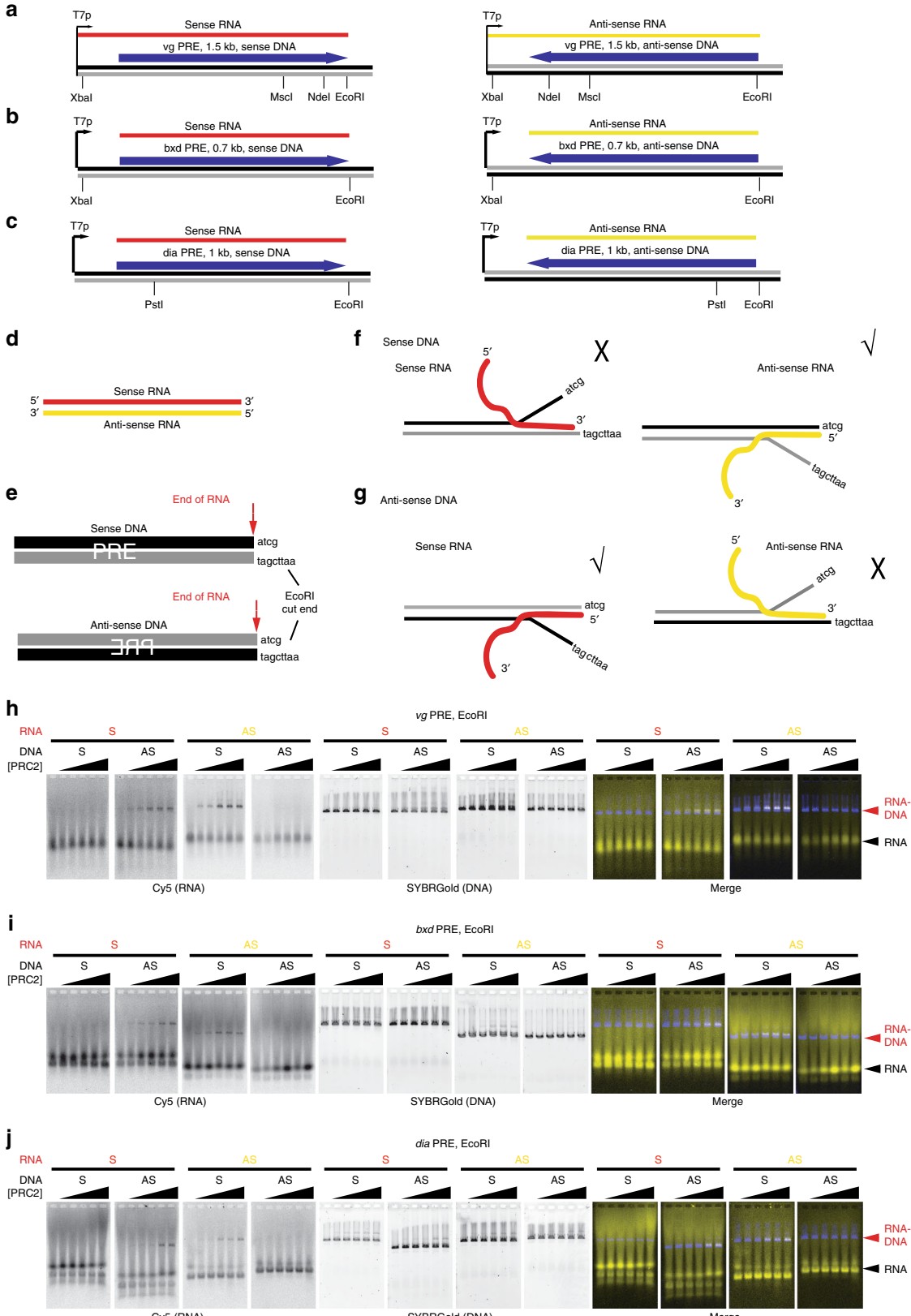

**Fig. 5 Pattern of RNA used by PRC2 on EcoRI-digested DNA templates. a–c** Schematic of DNA and RNA templates for the *vg* (**a**), *bxd* (**b**) and *dia* (**c**) PREs. DNA template were linearized at one of the indicated sites. **d, e** Schematic of RNA and DNA templates. RNA transcripts were produced from linear templates that end four bases before the EcoRI site so that the sequence indicated is not present in the RNAs. **f, g** Schematics of the four possible hybrids formed using the 2 RNAs and the sense (**f**) or anti-sense (**g**) DNA, indicating which two are observed. **h–j** show representative gels for each data set. Experiments were repeated three times, and all four sets for each PRE were always run in the same experiment. [PRC2] is 25–400 nM.

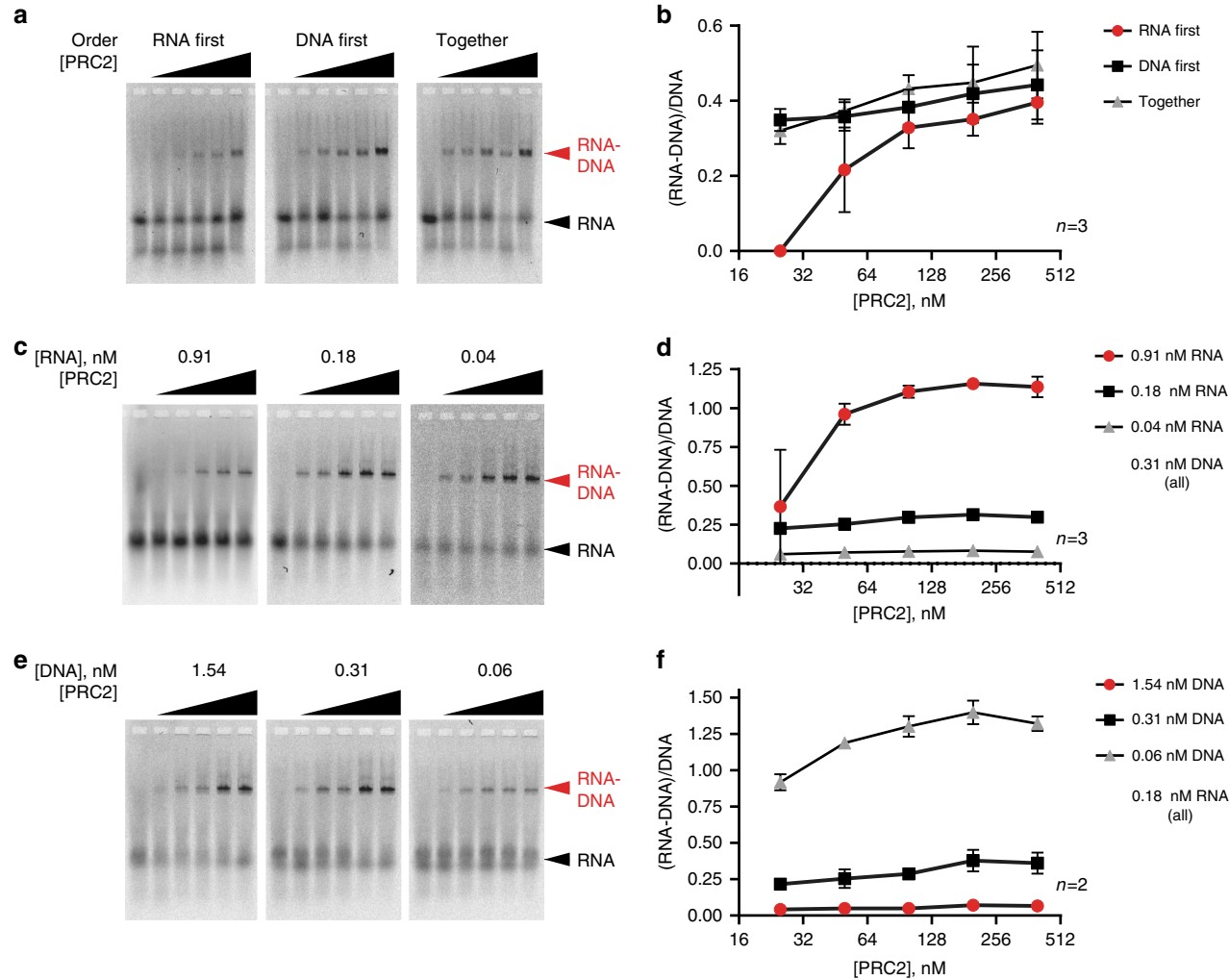

**Fig. 6 Characteristics of PRC2 strand exchange activity.** In (**a–f**), left panel shows a representative gel of Cy5-labelled RNA, and graph (right panel) summarizes multiple experiments. **a, b** Effect of order of addition of nucleic acids. **c–f** Effect of increasing RNA (**c, d**) or DNA (**e, f**). Asterisks: red = 0.91 nM vs. 0.04 nM; black = 0.91 nM vs. 0.18 nM. I- All PRC2 titrations are 25–400 nM. All graphs show the mean +/− S.E.M. Source data are provided in Source Data file.

depend on PRE transcription. Indeed, >70% of R-loops formed at PREs overlap an annotated coding or non-coding RNA, and PREs with R-loops are more likely to have RNA Pol II signal in ChIP-seq experiments. However, ~67% of PREs where we did not observe R-loops also overlap an annotated transcript. Further, a fraction of PREs with R-loops (and a fraction of total R-loops) either do not overlap any annotated transcripts, or overlap a transcript in the opposite orientation as the R-loop. While some of these discrepancies likely reflect incomplete annotation of rare transcripts, they raise the intriguing possibility that the RNA used to form the R-loops could be supplied in trans. Careful analysis of the RNA component of R-loops at PREs will be needed to resolve this. Although speculative at this time, the ability of PRC2 to induce RNA–DNA hybrids could contribute to non-co-transcriptional R-loop formation.

We find that PRC2 can induce RNA–DNA strand exchange from RNA and linear dsDNA in vitro. A small number of other proteins have been shown to have similar activity, using various types of substrates. These include the repair proteins Rad52/RecA[45–48] and PALB2[49], the human capping enzyme (CE)[50], the viral protein ICP8[51] and the telomere-inding protein TRF2[52]. Like the activity of PRC2, none of these reactions require ATP hydrolysis (although R-loop formation by RecA is stimulated by

ATPγS[45]), and most use linear DNA substrates[46,47,49,50] or an unpaired or ssDNA region[45,48]. The exceptions are TRF2 and ICP8. ICP8 can mediate R-loop formation from an RNA and a supercoiled plasmid[51]. TRF2 stimulates invasion of RNA oligos into a supercoiled plasmid encoding a telomeric DNA array[52], but the mechanism is believed to be induction of positive supercoiling by TRF2 that facilitates DNA unwinding and RNA invasion[53]. RNA–DNA strand exchange has been investigated most closely for Rad52, and its homologue RecA[45,48]. Rad52 has been shown both to carry out "inverse strand exchange" where Rad52 first binds the dsDNA, allowing RNA strand exchange[46], and to use an RNA-bridging mechanism, in which Rad52 first binds the RNA, and can bridge two dsDNA fragments by forming RNA–DNA hybrids with segments of each of them[47]. Both of these mechanisms are candidates to mediate RNA-mediated repair of DSBs[46,47]. PRC2 requires a DNA end for RNA–DNA strand exchange in vitro; for this activity to occur in vivo, either a DNA break would be required, or PRC2 would need to be able to use DNA opened by (an)other factors, or by transcription. These requirements may limit PRC2 strand exchange activity at PREs. In order to fully understand the impact of this activity in vivo and to what extent PRC2 contributes to R-loop formation at PREs, additional experiments will be necessary. Interestingly, Topoisomerase II

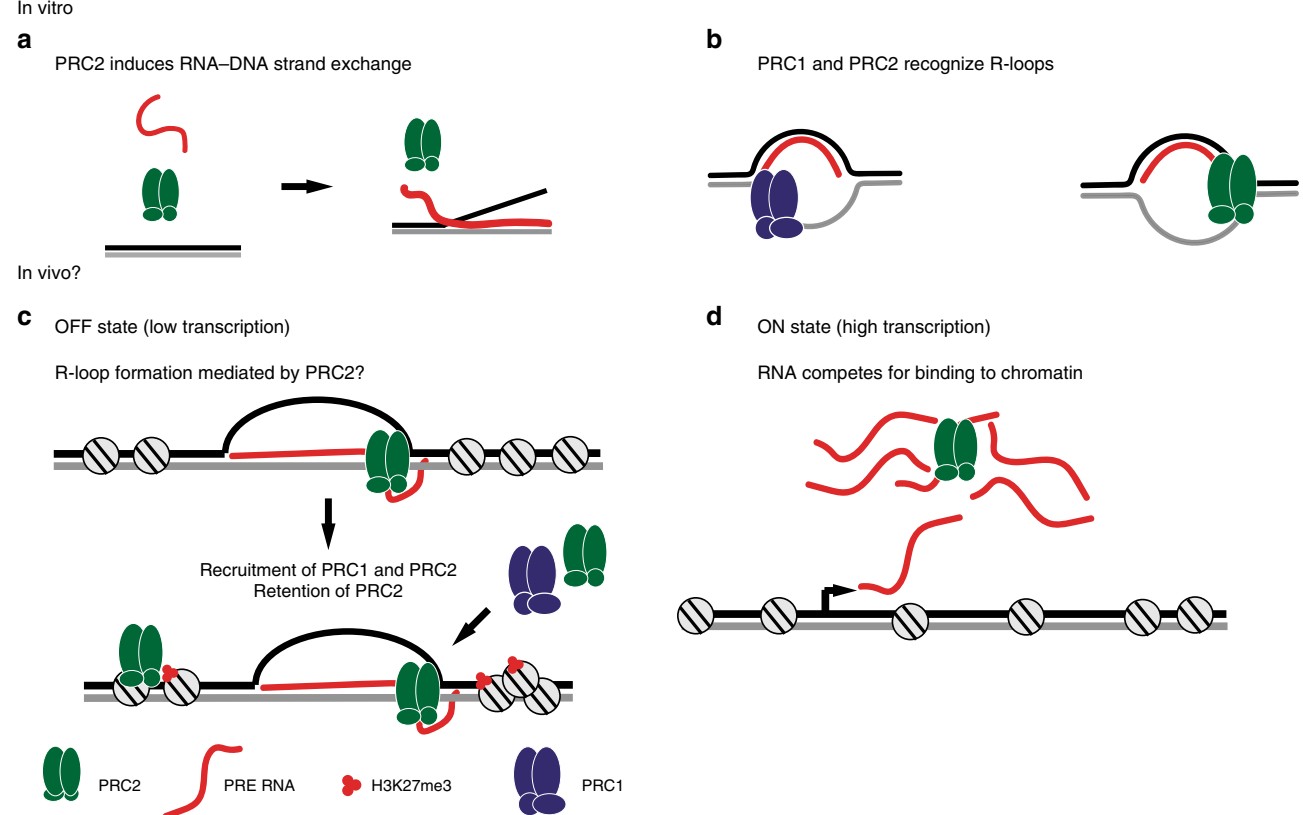

**Fig. 7 Model for the role of R-loop formation driven by PRC2 in PcG gene silencing.** Top panels (**a**, **b**) summarize our in vitro results with PRC2 (**a**, **b**) and PRC1 (**b**). Bottom panels (**c**, **d**) are a speculative model for how these activities could contribute to establishing or maintaining the OFF state at PREs. The ability of RNA to compete for PRC2 binding to chromatin is based on extensive observations in the literature. **a** PRC2 mediates RNA–DNA strand exchange, creating RNA–DNA hybrids. **b** PRC1 and PRC2 recognize R-loops. **c** Low levels of transcription through PREs could allow R-loop formation, which could depend on PRC2 activity (but could also be independent). R-loop formation sequesters the RNA and inhibits new RNA production, allowing PRC2 to be retained on chromatin. Recruitment of PRC1 and PRC2 (and other factors) may occur through recognition of the R-loop. **d** RNA produced by high levels of transcription competes for PRC2 binding to chromatin. See Discussion for details. This model is inspired by that of Ringrose[12], but incorporates our new in vitro observations.

interacts with a subunit of PRC1, colocalizes with PcG proteins in the BX-C, and is implicated in PRE-mediated silencing[54]; transient Topo II induced breaks have been implicated in regulation of transcription and chromatin compaction[55,56], and could also be used by PRC2. It is also possible that the activity of PRC2 contributes to RNA–DNA strand exchange at DNA breaks where RNA–DNA hybrids have been shown to form[57] and where PRC2 is recruited[58,59].

The connection between RNA and PRC2 has been recognized for some time, in species from plants to humans[11–13,60], but mechanisms beyond RNA binding by PRC2 have not previously been described. Our discovery of PRC2-mediated RNA–DNA strand exchange, suggests one mechanism to connect RNA to PcG targeting and function.

## Methods
**S2 cell culture**. *Drosophila* S2 cells were purchased from Invitrogen, and grown in Schneider's media (Invitrogen) with 10% heat inactivated, insect cell tested FBS (Invitrogen). Cells were cultured at 27 °C in suspension in shaking flasks.

**Drosophila collection**. Oregon R flies were grown at 25 °C. Embryos were collected on apple juice places and dechorionated for 2 min in 50% bleach before being washed with $H_2O$ and stored at −80 °C.

**Total nucleic acid extraction from S2 cells**. $8 \times 10^7$ S2 cells were washed with 1× PBS and resuspended in 10 mL TE. Cells were lysed O.N. at 37 °C in presence of 0.5% SDS and 62.5 µg/mL of proteinase K. After phenol-chloroform-isoamyl acohol extraction, total nucleic acids were precipitated in the presence of 0.3 M

sodium acetate pH 5.2 and 2.4 volume of 100% ethanol. Nucleic acids were washed carefully five times with 70% ethanol, and resuspended in TE.

**Total nucleic acid extraction from *Drosophila* embryos**. Total nucleic acids were extracted from 500 µL of Oregon R embryos as described in Ejsmont et al.[61] with the omission of RNaseA. After precipitation the nucleic acids were washed carefully five times with 70% ethanol, and resuspended in TE. This material was subsequently processed for DRIP analysis as described below.

**DRIP-seq and DRIP-qPCR**. The DRIP protocol was adapted from Ginno et al.[62]. Five hundred micrograms of total nucleic acid were divided in 3 and each treated with 150 µg of RNaseA in presence of 0.5 M NaCl for 3 h at 37 °C. gDNA was purified by phenol-chloroform extraction followed by ethanol precipitation and sonicated to 300 bp using a Covaris. Fragmented gDNA was treated with 2 U of RNaseIII[27] (Thermo Fisher) +/−10 µg each of homemade RNaseH I and RNaseH II overnight at 37 °C. Immunoprecipitation was performed as described in Ginno et al.[62]. After elution, samples were purified with a PCR clean-up column (Macherey-Nagel) with NTB buffer to get rid of SDS followed by a DNA clean and concentrator column (Zymo Research). For sequencing library preparation, material from three immunoprecipitations were pooled. Libraries were prepared using the NEB next Ultra II kit for a directional library for Illumina (NEB). For strand specific DNA sequencing of the RNA–DNA hybrids, we started at the second strand synthesis step and ligated with NEB-next multiplex oligonucleotides for Illumina (NEB). Paired-end sequencing was performed on an Illumina HiSeq 2500 at Genome Quebec.

For qPCR, input was diluted 10-fold and IPs twofold in water. PCR was carried out in 5 µl reactions consisting of 2 µl DNA, 2.5 µl PowerUp SYBR Green master mix (Thermo Fisher) and 0.25 µl of a 1 µM stock of each primer diluted in water. Standard curves were generated using a log titration of Drosophila genomic DNA purified from S2 cells (25 to 0.025 ng). Data were collected using a Viaa7 PCR system (Thermo Fisher) with 40 cycles. The standard curve was used to calculate

DNA amounts. All standard curves had $R^2$ values of 0.9 or higher. Oligonucleotides used for qPCR[32,63] are list listed in Supplementary Table 1.

**DRIP-seq analysis**. FastQ files of DRIP-seq reads were trimmed with Trimmomatic (PE –phred33), using the GenPpipes ChIP-seq pipeline (steps 1–3)[64]. Reads with both mate pairs were aligned to the dm3 version of the *Drosophila* genome using Bowtie2/2.3.1(–fr –no-mixed –no-unal)[65]. Sam files generated by Bowtie2 were converted to bam, sorted and indexed (samtools (v. 1.4.1)[66] and Picard (http://broadinstitute.github.io/picard) MarkDuplicates (default parameters) was used to remove duplicates. To generate strand specific bam files, samtools was used as follows:

Forward strand: samtools view –f 99; samtools view –f 147, followed by samtools merge.

Reverse strand: samtools view –f 83; samtools view –f 163, followed by samtools merge.

Peaks were called for DRIP versus input and DRIP versus RNaseH treated using MACS2[67] (v. 2.1.1) (-f BAMPE –bw 250 -g dm –mfold 10 30 -q 0.01). For strand specific peaks, strand specific files were used (e.g. F-strand DRIP, F-strand input, F-strand RNaseH). Peaks present in both DRIP vs. input and DRIP vs. RNaseH were retained (BEDTools intersect)[68] for each duplicate. Finally, BEDTools (intersect) was used to retain only peaks present in both duplicates, which were used for further analysis. The correlation between the replicates was examined using multiBigwigSummary on Galaxy (bin size: 1000 bp) followed by plotCorrelation using the Pearson correlation method. Correlations for replicates were: 2–6H 0.97, 10–14H 0.87, S2 0.99. Bigwig files were generated using DeepTools[69] v 2.5.3 (–binSize 10\ –normalizeUsingRPKM).

A list of PREs (Supplementary Table 2) was generated by combining predicted PREs[70], PcG binding sites conserved through *Drosophila* species[71], and additional PREs from recent reports[11,63,72]. Multiple PREs predicted in the repeated histone gene clusters were removed, although ChiP-seq peaks for PcG proteins are observed at these sites. Finally, overlapping or touching PREs were merged (using BEDTools). The list of genomic coordinates for PREs is in Supplementary Table 2.

To analyze overlaps between R-loops and PREs or other genomic elements, bed files of peak calls of unstranded, forward, and reverse strand peaks were merged to produce a consolidated set of R-loops. Overlap of R-loops or PREs with different genomic elements (Supplementary Fig. 1e–h) were generated with Pavis, with upstream and downstream regions both set at 5000 bp[73]. To correlate gene expression levels with R-loop formation (Supplementary Fig. 2a, b) the overlapping or closest gene to each R-loop was identified using BEDTools, ClosestBed on Galaxy. Level of gene expression were determined using RNA-seq data from embryos or S2 cells and genes were divided into categories based on their FPKM level (no to extremely low expression: FMPK < 1, low expression: 1<FPKM < 10, moderate expression: 10<FPKM < 50 and high expression: FPKM > 50). To compare R-loop orientation to annotated transcripts, the "all EST" track was downloaded from UCSC, and BEDTools was used (intersect intervals, only overlaps occurring on the same strand).

To analyze overlap of PREs with PcG protein, RNA Pol II, or H3K27me3 ChIP-seq peaks (Fig. 1d–f, Supplementary Fig. 2d–k), previously processed bed files were used with BEDTools (intersect). To analyze ChIP-seq signal intensity over PREs with and without R-loops, raw data (FASTQ files) were downloaded using the SRA tooolkit (v2.9.6) (http://ncbi.github.io/sra-tools/, SRA Toolkit Development Team), aligned with Bowtie2 as described above, duplicates removed (Picard), and RPKM-normalized bigwig files generated (DeepTools bamCoverage). BEDOPS[74] (v2.4.34) was used to convert bigwig files to wig and then bed files, and read densities quantified using BEDOPS bedmap (bedmap –count –echo-ref-name). Read densities over each PRE were divided by the PRE length to obtain the final values. All data sets used to analyze R-loops are listed in Supplementary Table 3.

To analyze overlap with annotated genes and RNA, we first converted DRIP-seq peaks to the dm6 genome using the UCSC genome browser liftOver tool. DRIP-seq peaks mapping to heterochromatin, ChrU, and ChrMT were removed. Unstranded DRIP-seq peaks were overlapped with either all ESTs ("mRNA and EST") or all genes ("Genes and Gene Predictions", UCSC Table Browser) using bedtools –intersect to produce the table shown in Extended Data Fig. 1c.

To analyze the overlap with annotated RNAs in a strand specific manner, we used the strand-specific peak calls. Files were prepared as described above. To remove peaks with R-loops called on both strands, we first intersected the F and R strand files, and removed DRIP-seq peaks that were called on both strands. The remaining peaks were overlapped with all ESTs using bedtools intersect with the –s or –S options to obtain overlaps with sense and anti-sense transcripts. The same process was carried out for PREs, except that PREs were first intersected with F and R DRIP-seq peaks. In the analysis shown in Extended Data Fig. 2c, "sense" indicates that the peak overlaps an annotated RNA in the sense orientation, but may also overlap an annotated RNA in the antisense orientation; "antisense" indicates overlap only with an antisense transcript.

**Protein expression and purification**. Human RNaseH1: A 6×-His tag was added to MBP-hRNaseH1, which was expressed in and purified from *E. coli* based on a previously described protocol[75,76], except that Ni-NTA beads were used for the first step instead of amylose beads.

hRNaseH2: The RNaseH2 trimer was produced using the multi-cistronic pMAR22 vector essentially as described[76].

PRC1ΔPh, PRC2, dSxc, hNFY: PRC1ΔPh and PRC2 were expressed in and purified from Sf9 cells, with the following modifications to previously published protocols for anti-FLAG affinity purification[77–79]. For PRC1ΔPh, nuclear extracts were prepared from Sf9 cells infected with viruses for the 4 subunits[79] but nuclei were purified through a sucrose cushion prior to nuclear extraction. During the purification, the 2 M KCl wash in the published protocol was replaced with a wash consisting of BC2000N + 1 M Urea (20 mM Hepes, pH 7.9, 2 0.4 mM EDTA, 2 M KCl, 1 M deionized urea, 0.05% NP40, no glycerol). Additionally, prior to eluting the protein, anti-FLAG beads were incubated 3–5 volumes of BC300N with 4 mM ATP + 4 mM MgCl$_2$ for 30 min. at room temperature. This step reduces the amount of HSC-70 that co-purifies with PRC1ΔPh.

For PRC2 expression and purification, E(Z) was tagged with 6-His, and either Esc or Su(Z)12 with FLAG, and baculovirus infected Sf9 cells were harvested after 3 days. PRC2 was purified by anti-FLAG affinity as described[78] followed by Ni-NTA. FLAG peptide elutions were carried out in BC300 without EDTA or DTT. FLAG elutions were passed over Ni-NTA beads twice, beads were washed with 30 volumes of BC300 (without EDTA or DTT) and eluted in BC300 + 250 mM Imidazole. Eluted protein was pooled and dialyzed through three changes of BC300 with EDTA, PMSF, and DTT. PRC2 was concentrated to ~1 mg/ml, NP40 was added to 0.05%, and protein was stored at −80 °C.

Extract preparation and anti-Flag purification of F-Sxc and F-NFY were as described for PRC2.

For glycerol gradient fractionation of PRC1ΔPh or PRC2 (Supplementary Fig. 4e, f), 5–10 µg of protein were loaded on a 280 µl step gradient (35/30/25/20/15/10/5% glycerol) in BC300 buffer. Gradients were centrifuged for 3 h at 367,600 g at 4 °C using an SW55Ti rotor and resolved into 50 µl fractions.

**Oligonucleotides assembly and labelling**. DNA and RNA oligonucleotides described in Supplementary Table 1 were diluted to 1 µM in TE supplemented with 50 mM NaCl (TE-50), boiled and cooled O.N. with the exception of ssDNA which was snap frozen. All substrates were gel purified on 8% acrylamide 0.5X TBE gels and nucleic acids were extracted O.N. by incubated the band of interest in 20 mM Tris-HCl pH 7.5, 150 mM NaCl, 0.1% SDS, 10 mM EDTA. Nucleic acids were precipitated with ethanol, washed with 70% ethanol and resuspended in TE-50. Substrates were labelled with T4 PNK (New England Biolabs, NEB) and $^{32}$P-γATP (PerkinElmer). Samples were purified by phenol-chloroform extraction, followed by purification through a G-25 spin column equilibrated with TE-50.

**Filter binding assay**. Filter binding was carried out essentially as described[80], using a nitrocellulose filter (to capture protein-nucleic acid complexes) (Biotrace) stacked on a charged nylon membrane (to capture free nucleic acids) (HYBOND membrane, GE Healthcare) with a slot blot apparatus. Nitrocellulose filters were prepared by incubating in 0.4 M KOH for 10 min, washing extensively with H$_2$O, and equilibrating at least one hour in binding buffer. Nylon filters were equilibrated at least 10 min. in 0.4 M Tris, pH 8.0. We first measured the active concentration of two preparations of PRC1ΔPh; concentrations reported in Fig. 2 are the active concentration. To measure binding to oligonucleotide substrates, PRC1ΔPh and PRC2 were titrated into reactions with 0.01 nM of DNA bubble, R-loop or dsDNA in 20 µl reactions containing 12 mM Hepes, pH 7.9, 0.12 mM EDTA, 120 nM KCl, 1 mM DTT, 0.01% NP40, and 12% glycerol and incubated 30 min. at 30 °C. To apply reactions to the filters, each well was washed with 100 µl binding buffer, sample was applied, and wells were washed twice with 100 µl of binding buffer. Membranes were exposed to a phosphor imager screen, scanned on a Typhoon imager (GE Healthcare), and quantified with ImageQuant.

**EMSA**. PRC1ΔPh and PRC2 were titrated into reactions with 0.01 nM of DNA bubble, R-loop or dsDNA in the same reaction conditions as for filter binding except that 50 ng/µL BSA were included Reactions were resolved on 6% acrylamide 0.5 × TBE gel. Gels were dried and exposed to a phosphor imager screen and scanned on a Typhoon imager.

**PRC1ΔPh and PRC2 incubation with filter binding probes**. PRC1ΔPh and PRC2 were titrated into reactions with 20 nM of DNA bubble, R-loop, dsDNA, ssDNA and RNA in the same conditions as for filter binding. Proteins were digested with 3 µL of DSB-PK (6.7 µg/µL of proteinase K (Biobasic), 1% SDS, 50 mM Tris-HCl pH 8.0, 25% glycerol and 100 mM EDTA) for 30 min at 50 °C, nucleic acids were resolved on 8% acrylamide 0.5× TBE gels and stained with SYBRGold (Thermo Fisher).

**Plasmids for strand exchange**. PRE sequences were amplified by PCR from *Drosophila* genomic DNA and cloned into the pET-Blue1 vector (Millipore) downstream of the T7 promoter. Detailed maps are available on request. For strand exchange, plasmids were digested with a single restriction enzyme, purified by phenol-chloroform extraction, and ethanol precipitated.

**RNA production and labelling**. RNAs were produced from linear templates using the Ampliscribe T7-flash transcription kit (Lucigen) using the manufacturer's protocol in the presence of 25 mM of amino-allyl UTP (Sigma). After purification, RNAs were labelled with NHS-Cyanin-5 (Kerafast) in 70 mM NaHCO$_3$ pH 8.8 with murine RNase inhibitor (NEB) for 2 h at RT. RNAs were then precipitated with 0.3 M sodium acetate pH 5.3, glycogen and ethanol, washed with 70% ethanol, and resuspended in TE. RNAs were passed through a G50 column equilibrated with TE. The quality of labelled RNA and efficient removal of free dye were determined by loading the RNA on agarose gels.

Radiolabelled RNAs were produced from circular templates by transcribing 600 ng of DNA in RNA polymerase buffer (NEB), 1 mM DTT, 625 μM of rNTP (NEB), 6.5 nmol of radiolabelled UTP, 200 U of RNase inhibitor, and 250 U of T7 or T3 RNA polymerases (NEB) in 100 μL O.N. at 37 °C. DNA was removed from the reaction by adding 4 U of DNaseI (NEB) and incubating 2 h at 37 °C. RNAs were extracted with phenol-chloroform, ethanol precipitated, washed with 70% ethanol, resuspended in TE, and stored at −20 °C.

**RNA strand exchange assay**. PRC2, diluted in BC300N was incubated with the indicated amount of DNA and fluorescent- or radio- labelled RNA for 25 min. at 30 °C in 180 mM KCl, 5 mM MgCl$_2$, 1 mM DTT and with 50 ng/ μL BSA in 10 μL reaction. After incubation, samples were treated with 3 μL of DSB-PK for 30 min. at 50 °C and resolved on 0.8% agarose 0.5X TBE gel. Gels were stained with SYBR-Gold or ethidium bromide, and imaged on a Typhoon Imager. For experiments with radio-labelled RNA, gels were transferred to HYBOND membrane and exposed to a phosphor imager screen.

For nuclease treatment of strand invasion products without phenol-chloroform extraction, after incubation with PRC2, samples were treated immediately with nucleases. For RNaseH treatment, 10× RNaseH buffer was added to a final concentration of 1×, followed by 2.5 U (radio-labelled RNA) or 1.25 U (fluorescently labelled RNA) of RNaseH. For RNaseA treatment, reactions were supplemented with 500 mM NaCl and 50 pg of RNaseA (Qiagen) were added. Reactions were incubated for 30 min. at 30 °C. For phenol-chloroform extracted samples, reactions were stopped with 3 μL of DSB-PK and incubated for 30 min at 50 °C. Nucleic acids were extracted with phenol-chloroform followed by ethanol precipitation and resuspension in TE. Nuclease digestion was carried out as described above. Nuclease digestions were stopped by the addition of 3 μL of DSB-PK, and samples were incubated 30 min at 50 °C before analyzing on agarose gels. When the order of DNA and RNA addition was tested, the first nucleic acid was added to PRC2 for 10 min at 30 °C before the addition of the second.

For the pre-incubation assay shown in Fig. 4d–f, DNA was incubated with PRC2 under conditions described above, proteins were removed by digesting with DSB-PK O.N. at 50 °C. DNA was purified using a PCR clean-up column (Macherey-Nagel)and eluted in 10 mM Tris. This DNA was used in RNA strand exchange reactions.

**RNA strand exchange after apyrase treatment**. Ten micrograms of PRC2 and 330 ng of Cy5-labelled RNA were incubated 30 min. at 30 °C with respectively 0.5 and 1 unit of apyrase (NEB) in 15 or 30 μL reaction. Apyrase was remove by passing PRC2 and RNA on G50 column equilibrated respectively with BC300N and TE. PRC2 and RNA treated with apyrase were used in strand exchange reactions.

**RNA strand exchange followed by DRIP in vitro**. DNA, RNA and PRC2 was incubated as describe for RNA–DNA strand exchange in 100 uL reactions. Reactions were stopped with 30 uL of DSB-PK and digested for 30 min. at 50 °C. Nucleic acids were extracted with phenol-chloroform, ethanol precipitated, washed with 70% ethanol and resuspended in TE. RNA–DNA hybrid containing fragment were immunoprecipitated by incubated O.N. at 4 °C with protein G beads pre-incubated with 2.5 μg S9.6 antibody (Kerafast ENH001 and M. Wilson) and competitor DNA (pUC19 plasmid digested with DrdI, AlwNI and ScaI-HF). Beads were washed three times with 10 mM NaPO4 pH 7.0, 140 mM NaCl and 0.05% Triton X-100. Nucleic acids were eluted from the beads by incubating them 45 min at 50 °C in presence of 50 mM Tris pH 8.0, 10 mM EDTA, 0.5% SDS and 56 μg proteinase K. Samples were resolved on 0.8% agarose 0.5× TBE gels. Gels were stained with SYBRGold and imaged on a Typhoon Imager.

**Oligonucleotide annealing assay**. Phosphorylated ssDNA or RNA oligos corresponding to a sequence in the bxd PRE (Supplementary Table 1) were used at a final concentration of 40 nM. One ssDNA oligo is 5' labelled with Cy5. Annealing was carried out with the same reaction conditions as RNA strand exchange except that no MgCl$_2$ was added and the [KCl] was 60 mM. Reactions were incubated for 25 min at 15 °C, and stopped by adding 200 nM of unlabelled ssDNA oligonucleotides and 25 ng of *vg* RNA and incubation for 10 min at 15 °C. Reactions were loaded on 8% acrylamide 0.5× TBE gels. Gels were stained with SYBRGold and imaged for Cy5 and SYBRGold on a Typhoon Imager.

**Nuclease activity assay on oligonucleotides**. Oligonucleotides were phosphorylated with T4 PNK (NEB). dsDNA oligonucleotides were annealed by incubating equal amounts of ssDNA in TE-50, boiling for 5 min. and slow cooling over several hours. ssDNA oligonucleotides were boiled and transferred immediately to ice. PRC2, T7 endonuclease (NEB), Exonuclease I (NEB), λ exonuclease (NEB) or Exonuclease III (NEB) were incubated with phosphorylated ssDNA or dsDNA oligonucleotides under RNA–DNA hybrid forming conditions. Reactions were stopped by adding DSB-PK and incubated 1 h at 50 °C. Samples were denatured by addition of 26% formamide, 0.3 mM EDTA, 3.3 mM NaOH and boiled before loading on denaturing gels (10% acrylamide, 1× TBE, 7 M urea). Gels were stained with SYBRGold and imaged on Typhoon Imager.

**Gel quantification**. For quantification of DNA and RNA–DNA hybrids from phoshpor imager and SYBR gold scans using ImageQuant, RNaseA-treated samples were used. In cases where gel flaws obscured quantification of a lane, the gradient was excluded from analysis. Background subtraction was done using the rolling ball method. For band selection, the smallest possible "fixed width" bands that capture the whole signal were set for each gradient. These bands were placed in each lane so that every fraction was quantified. The signal from the bottom three fractions was divided by that for the total of the gradient for the fraction bound.

RNA strand exchange assay gels of Cy5-labelled RNA were imaged on Typhoon Imager (GE Healthcare) were quantified using ImageQuant (GE Healthcare). Lanes were created manually, then background was removed using minimum profile method and bands were identified manually.

**Electron microscopy**. Two hundred nanograms of PRC2 or 20 units of exonuclease III were incubated with 1.9 nM of pFC53 DNA linearized with HindIII using RNA–DNA hybrid formation conditions. Reactions were stopped with 15 μL of DSB-PK and incubated O.N. at 50 °C. DNA was purified on PCR clean-up columns (Macherey-Nagel) with NTB buffer. DNA was incubated with *E. coli* SSB (NEB) at a ratio of 6 μg SSB per μg DNA on ice for 10 min[81]. Samples were purified through a G-50 column equilibrated in TE. Samples were diluted 1:5 in 2 mM Spermidine, 150 mM NaCl, 1 mM MgCl$_2$, and applied to glow-discharged 400-mesh grids coated with thin carbon (#Cu-400CN, Pacific Grid Tech) for 5 min. Grids were washed through two drops of 100 mM MgOAc, and stained with 3 drops of 2% Uranyl Acetate (Electron Microscopy Sciences), blotted and air-dried. Grids were photographed at 120 kV using an FEI Technai G2 Spirit BioTwin Cryo-TEM at the McGill Facility for Electron Microscopy Research. Note that both positively and negatively stained regions were observed on the grids; positive staining is shown in Fig. 4 because it facilitates visualization of SSB-coated DNA filaments.

**Statistics and curve fitting**. Graphpad Prism was used for statistics and curve fitting. For time course data, the equation $Y = ABmax*(1−e^{−k*X})$ was used; for binding data, $Y = (ABmax*X)/(X + K_d)+b$ was used. Fisher's exact test was used to compare curve fits. For students *t*-tests (Fig. 2), we used Holm-Sidak correction for multiple comparisons, alpha = 0.05, with all points assumed to come from populations with the same S.D. For comparing distribution of ChIP-seq peaks or transcripts (Supplementary Fig. 2), data were organized as a contingency table (i.e. columns = peak/no peak, rows = R-loop/no R-loop) and compared using Fisher's exact test, reporting two-sided *P*-values. To compare ChIP-seq read intensities over PREs, Mann–Whitney tests were used, with two-tailed *p*-values reported.

RegioneR[82] was used to conduct permutation tests of the overlaps between PREs and R-loops, or ChIP-seq peaks and PREs with and without R-loops (1000 permutations, randominze.function = randomizeRegions, evaluate.function = numOverlaps, count.once = TRUE, genome = "dm3").

**Reporting summary**. Further information on research design is available in the Nature Research Reporting Summary linked to this article.

## Data availability
Sequence (DRIP-seq) data that support the findings of this study have been deposited in NCBI GEO with the accession code GSE127329. Other data that support the findings of this study are available from the corresponding author upon reasonable request. The source data underlying Figs. 2a–d, 3g, 4f, 6b, d, f and Supplementary Figs. 1b, 6d, 7b–c are provided as a Source Data file.

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

## Acknowledgements

We thank K. Sears at the McGill Facility for Electron Microscopy for assistance with sample preparation, E. Lécuyer's lab for assistance collecting *Drosophila* embryos, O. Neyret for advice on preparation of NGS samples, J. Mallette for technical assistance, C. Gentile for advice on bioinformatics, M. Wilson for S9.6 antibody, M. Reijns for plasmid to express hRNaseH2, M. Drolet for intellectual support, J.-Y. Masson for suggesting the EM experiment, Y-S Kang for help with bioinformatics, and F. Robert, M. Drolet, and members of the Francis lab for comments on the manuscript. This research was enabled in part by support provided by Calcul Québec (www.calculquebec.ca) and Compute Canada (www.computecanada.ca). Work in the N.J.F. lab was funded by CIHR 311557, in the K.J.A. lab by a grant from the David and Lucile Packard Foundation, NIH 5R01GM115882-03, and 5T32HL007151-40 (to D.G.), and in the F.C. lab by NIH R01-GM120607.

## Author contributions

Conceptualization: N.J.F. and C.A.; Investigation: C.A., O.A. and N.J.F.; Formal analysis: C.A. and N.J.F.; Resources: V.C., E.L.B., D.G., K.J.A., L.A.S. and F.C.; Writing: N.J.F., C.A., K.J.A. and F.C., Supervision: N.J.F., Funding acquisition: N.J.F., F.C. and K.J.A.

## Competing interests

The authors declare no competing interests.
