## [Peer Review File · Nature Communications]

Reviewers' comments:

Reviewer #1 (Remarks to the Author):

Alecki et al provide additional evidence for the association of polycomb group (PcG) proteins with R-loops in vivo and, more importantly, are placing a new model to explain how PRC2 facilitates the formation of R-loops through an RNA strand-invasion activity. Although one could argue that the latter part is not supported by experiments in cells, it is yet still a significant finding. Specifically, the link between PcG proteins to R-loops was established in previous publications, predominantly using cell-based model systems, while mechanistic studies are still lagging given the complexity of this system in vivo. Therefore, reconstituted systems in vitro are an excellent tool in order to address this specific problem and have the potential to reveal how R-loops regulate PcG proteins and vice versa.

While the actual observation of PRC2-mediated strand invasion is highly significant and would certainly justify a publication in this journal, extensive work needs to be done in order to exclude potential artifacts. Next, there some editing work that better be done by the authors, to avoid overstatements and limit speculations to what can be reasonably supported by the data.

MAJOR POINTS

1. Nuclease activity in protein preps could have led to the artificial observation of strand invasion: If small amount of DNA exonuclease activity is associated with recombinant PRC2 proteins (e.g. if carried from the purification process), it could digest all or part of one strand of the linear dsDNA probe that was assayed in Fig 3 and will thus expose the other single strand of that DNA. Next, the newly formed ssDNA will be available to bind RNA and give rise to an artificial observation of strand invasion in vitro. This is a serious concern given three independent observations that were made by the authors: (1) In Extended Data Fig. 5c, the DNA is hardly seen in the presence of high PRC2 concentration points, which is consistent with nuclease activity. (2) The observation of strand invasion is dependent on Mg²⁺ (Extended Data Fig. 6c), which is also required for the activity of many nucleases. (3) The authors observed very little strand invasion activity of PRC2 toward a circular plasmid (Extended Data Fig. 5g), which is not as susceptible to exonuclease activity as the linearized DNA that was used in most of the assays. Since strand invasion is a major point in this paper, the authors should do the minimum in order to exclude DNA exonuclease activity in their protein preps. Specifically, the authors should incubate all the PRC2 complexes that were used in this study (fly PRC2 and human PRC2-EZH1 and PRC2-EZH2) with low nanomolar concentration of either fluorescently-labeled or radiolabelled short dsDNA probe of approximately 100-200 bp, where exonuclease activity can be detected by electrophoresis (e.g. on acrylamide gels). A positive control should be included, e.g. an exonuclease from a commercial source, to demonstrate that the assay can detect exonuclease activity if present. Nuclease assays should be carried under the same experimental conditions (protein concentration, buffer, temperature, time etc.) as used in assays within Fig 3d-e, albeit without RNA. For complementation, the authors should repeat experiments in Fig 3d-e, where PRC2 will be replaced by a commercial DNA exonuclease that will yield

an ssDNA and dNTPs as products. If strand invasion is really mediated by PRC2, the results should ideally show that PRC2 prep is devoid of nuclease activity and, independently, that exonuclease activity cannot lead to the same observation as in Fig 3d-e.

2. Ideally, strand invasion should be demonstrated using a non-linearized template: In Fig 3, the author used a linearized plasmid to demonstrate efficient strand invasion, but it is concerning that strand invasion experiments using non-linear plasmids (Extended Data Fig. 5g) showed poor strand invasion activity. The authors should quantify the variations between strand invasion to linearize vs circular plasmids and comment on the potential reason(s) for these variations. If the authors have a concern regarding plasmid supercoiling as a potential negative factor for strand invasion, they can also compare non-linearized plasmid to non-linearized bacmid. If authors are concerned regarding another potential biophysical/biochemical restraint then they should test them experimentally. Leaving this observation without interpretation and testing is inappropriate, as one could expect strand invasion to take place also in the context of a circular dsDNA contract.

3. Binding assays for PRC1, as done, cannot support for variations in affinity: The authors state that R-loops are recognized by the PcG complex PRC1 in vitro. Yet, the binding “curve” for PRC1 in Fig 2e is actually linear, not a sigmoidal binding curve (an example for a good-looking sigmoidal binding curve is in Fig2f, where the same assay carried out for PRC2). A linear binding dependency, between protein concentration to fraction-bound nucleic acid, represents a titration experiment, which typically occurs where the probe (DNA) concentration is close to K_d concentration. In a titration experiment, the slope of the line in the plot does not represent affinity, but rather binding stoichiometry and/or the fraction of protein that is active in nucleic acid binding, e.g. see Ryder et al. 2008 (PMID: 18982286). Based on the information provided by the authors, DNA concentration seems to be around 2 nM, which is very close to the PRC1 concentration that was assayed: 5-20 nM. The problem likely occurs also in Extended Data Fig 5d. In order to assess variations in affinity, even qualitatively, the authors need to repeat the experiment in Fig 2e under conditions that will provide a sigmoidal binding curve, not a linear titration. This can be done by significantly reducing the probe concentration below K_d or (less optimal) to increase the salt concentration in order to increase K_d high above the probe concentration.

4. The models presented in the model figures are not supported by the data: The authors do not provide any direct evidence to link the strand invasion activity that was observed in vitro to the function of PRC2 in cells. Given the complexity of the system and what is known so far, it is completely understandable if the authors would prefer to leave a comprehensive analysis in vivo to the scope of future studies. Yet, they should at least remove the model in Extended Data Fig. 10, which is completely imaginary, might be either right or wrong and is certainly not supported by the data, which did not bring into consideration the act of transcription, nucleosomes, epigenetic marks or anything else from this biological system in cells (with the exception of a basic association and correlative analyses in Fig 1). Furthermore, the authors should simplify the model presented in Extended Data Fig. 9. Specifically, although the three models proposed in Extended Data Fig 9a-c can all be tested biophysically in vitro, the authors didn't do it. Assuming the authors will be able to address concerns brought up above, the only model that is supported by the data is that PRC2 facilitates the formation of strand invasion. This is

a significant finding, but lack mechanistic details and can, therefore, be represented in a simple one-step model.

MINOR POINTS

5. The last sentence in the abstract state "... our findings suggest formation and recognition of non-canonical nucleic acid structures as an epigenetic mechanism". As exciting this idea is, it is a long stretch beyond the data. Supporting this statement will require the authors to perform experiments in cells to demonstrate that the inheritance of the repressed state through mitosis is dependent on the formation of PRC2-dependent R-loops and R-loop-dependent recruitment of PRC1. The authors can decide if they wish to perform these experiments or to revise that statement, but it is strongly recommended that they will do at least one of these.

6. Extended Data Fig. 1c: the table state "101460/17149 (61%)", but the math seems wrong. Is there a typo in "101460"?

7. Based on experiments in Fig 4e-f, the authors stated that RNA inhibits R-loop formation, but the data argue otherwise. Specifically, if instead of presenting (RNA-DNA)/(total RNA), in Fig 4f, the authors will present (RNA-DNA)/(total DNA) this will be evident.

8. The corresponding author has a solid and unquestionable experience from previous studies in the purification of PRC1. Yet, the purification of recombinant PRC1 in this study was modified and done under harsh conditions, with buffer containing 2 M KCl and 1 M Urea used for extraction from insect cells (Sf9) nuclei. The authors should provide evidence for complex assembly and solubility (ideally a gel filtration chromatography trace) and some evidence for enzymatic activity.

9. In Fig 2C, the total amount of RNA seen on the gel is reduced as PRC1 concentration increased. Can the authors exclude RNase contamination that is associated with their PRC1 protein prep? Or is there any other explanation for that?

Reviewer #2 (Remarks to the Author):

In this manuscript, Alecki and colleagues investigate the possibility that R-loops might participate in the repressive functions of PcG proteins in *Drosophila*. They propose (i) that R-loops form at a subset of PREs sequences *in vivo*, (ii) that PRC1 binds to R-loops *in vitro* and (iii) that PRC2 is able to create R-loops *in trans* using a possible strand invasion activity. They investigate whether this strand invasion activity is conserved in human PRC2 proteins. The biological significance of this putative strand invasion activity was not investigated. Together, their data also seem to suggest that R-loops could form at PREs both co-transcriptionally (Figure 2) and *in trans* through the strand-invasion activity of PRC2 (Figure 3&4). The

possible interplay between these putative -cis and -trans R-loops was neither investigated nor discussed. To synthesize their observations, the authors proposed that PRC2 creates R-loops in trans at PREs that are then recognized by PRC1. Whether the putative strand invasion activity of PRC2 is restricted to PREs sequences was neither explored nor discussed. The manuscript is relatively hard to follow and the starting hypothesis was not clearly explained. The figures would benefit from better labelling (especially when it comes to the hugely variable DNA and RNA concentrations used in the different assays). More importantly, the data lack important experimental controls. Overall, the authors failed to make a compelling case for their model.

Major points:

1. It is surprising that so many R-loops would form over “repressed” transcription units. At PREs, whether these R-loops are co-transcriptional or formed in “trans” by the putative strand invasion activity of PRC2, these loci must be transcribed at some point. A comparison of DRIP-seq maps to RNAP2 ChIP-seq maps would therefore have been more appropriate than a comparison to RNA-seq data as presented here, because RNA-seq only measures steady-state levels of soluble RNAs. Published RNAP2 ChIP-seq data might be available.
2. The demonstration that co-transcriptional R-loops form at PREs in vitro is not compelling.
 - a. One wonders how the authors can claim that their RNA-DNA hybrids are resistant to RNase A treatment when their data clearly show that their hybrid signal is sensitive to increasing RNase A concentrations (Figure 2b, Figure 3hi). Did the authors use sufficiently high salt concentrations in their reactions to limit the RNase H activity of RNase A?
 - b. The radiolabelled RNA seems to associate with both the supercoiled and the open form of the plasmid (Figure 2b, SC and R/N forms should be labelled on the figure). This makes little sense, as hybrid formation is expected to relax the plasmid. The authors justify this by saying that R-loops are probably small (“a few hundred bp”, legend of the Extended data Fig. 4) but “a few hundred bp” seem already to be quite big for R-loops (R-loops of around 200 bp were largely sufficient to completely relax pFC53 according to a recent publication by the Chédin lab). This might suggest that R-loops formed at PREs are indeed very small. It would therefore be nice to repeat some of their experiments with well-described R-loops of reasonable sizes, such as those formed in pFC53 (see below).
 - c. Their vg-PRE insert has been cloned 200 bp downstream of the T7 promoter according to the legend of Extended data Figure 6. The authors should make sure that this 200 bp sequence cannot contribute to the weak R-loop formation that they detected in their in vitro transcription assay, for example by using a T7-containing empty vector as negative control.
 - d. The authors should use dot blots with the S9.6 antibody to independently and more precisely quantify R-loop formation at vg-PRE and Airn.
3. In Figure 2ef, what does “DNA” mean? Has the “DNA” been transcribed? Or does it correspond to the mix of plasmids with and without hybrids after transcription? In the binding assays (Figure 2cd), a non-transcribed plasmid and/or a RNase H-treated plasmid must be used as specificity controls (this should really be the “DNA” in figure 2ef). It looks as if PRC1 associates preferentially with the relaxed form of the plasmid: is it because R-loops are longer on the relaxed form of the plasmid (the plasmid has been relaxed by the long R-loops), or is it simply that PRC1, contrary to PRC2, has a greater affinity for relaxed

than for supercoiled DNA? If the latter were true, the apparent preferential binding of PRC1 to “hybrids” would only be an indirect consequence of R-loop driven plasmid relaxation. This could easily be assessed with nicking enzymes. However, a better experiment would be to cut the plasmid in half after transcription (topology would not be an issue anymore): if the authors could show that PRC1 has indeed a stronger affinity for the R-loop containing fragment than for the plasmid backbone, the demonstration that PRC1 binds to R-loops would be a lot more compelling (this could be quantified by qPCR). As it stands, the demonstration that PRC1 binds to R-loops is not convincing. To strengthen their case even further, the authors should also assess the binding of PRC1 and PRC2 to well-described R-loops such as those formed at Airn.

4. The experiment showing that PRC2 promotes R-loop formation (Extended Data Figure 5g) should be quantified and should be confirmed using the S9.6 antibody (IP followed by qPCR). Could PRC2 really do this without the need for ATP?

5. The different “strand invasion” reactions (Figures 3&4 and Extended data Figure 6&7) are confusing because (i) the concentrations of DNA and RNA vary widely from one experiment to the next; (ii) it is surprising that PRC2 would be able to promote strand invasion without energy (no ATP was added to the reactions?) and (iii) the efficiency of the strand invasion reaction appears to be very weak and even weaker in human proteins. Even when the authors used only minute amount of RNA (fmol, Figure 4), a very large excess of PRC2 was not able to “recombine” the totality of that RNA (max 60%), even when the DNA concentration was increased to provide more templates to “recombine” with. One therefore wonders at the biological significance of these observations. In addition, Figures 3g/4d/S6d indicate that above ~30 nM of PRC2, the reaction seems to reach a plateau. Yet, the authors have made most of their measurements (Figures 4) at concentrations of PRC2 higher than 30 nM and they might have missed important observations as a result. The specificity controls that the authors used (Sxc and NFY) have a much lower affinity for RNA than PRC2 according to their own measurements. It would be more appropriate to use as specificity control a protein that has a comparable affinity for RNA, such as PRC1, which the authors have produced. Finally, how do the authors explain that the RNA that was not “recombined” with the DNA template did not associate with the massive excess of PRC2 considering the great affinity of PRC2 for RNA (no shift of the remaining RNA to the top of the gel like in their other EMSA assay (Extended data Figure 5b))?

Minor comments:

1. A better explanation of the starting hypothesis would be helpful.
2. The Discussion is very limited and should better mention the work recently published by Konstantina Skourti-Stathaki and Ana Pombo in *Molecular Cell*.
3. The idea that non-canonical nucleic acid structures, and especially R-loops, could convey epigenetic regulation has already been suggested by the work of Andrés Aguilera.

We appreciate the reviewers' detailed and thoughtful critique of our paper, and recognize that substantial issues were raised. We have done extensive experimental work to address these issues, which we feel has strengthened the manuscript. We have substantially revised the manuscript, both to include the additional data, and to address comments on the text. We have also removed data that we were not able to substantiate. As detailed in our response to point 2 from Reviewer 1, we have also replaced the term "strand invasion" with "RNA-DNA strand exchange", which we believe is a more precise description.

Reviewers' comments:

Reviewer #1 (Remarks to the Author):

Alecki et al provide additional evidence for the association of polycomb group (PcG) proteins with R-loops in vivo and, more importantly, are placing a new model to explain how PRC2 facilitates the formation of R-loops through an RNA strand-invasion activity. Although one could argue that the latter part is not supported by experiments in cells, it is yet still a significant finding. Specifically, the link between PcG proteins to R-loops was established in previous publications, predominantly using cell-based model systems, while mechanistic studies are still lagging given the complexity of this system in vivo. Therefore, reconstituted systems in vitro are an excellent tool in order to address this specific problem and have the potential to reveal how R-loops regulate PcG proteins and vice versa.

While the actual observation of PRC2-mediated strand invasion is highly significant and would certainly justify a publication in this journal, extensive work needs to be done in order to exclude potential artifacts. Next, there some editing work that better be done by the authors, to avoid overstatements and limit speculations to what can be reasonably supported by the data.

MAJOR POINTS

1. Nuclease activity in protein preps could have led to the artificial observation of strand invasion: If small amount of DNA exonuclease activity is associated with recombinant PRC2 proteins (e.g. if carried from the purification process), it could digest all or part of one strand of the linear dsDNA probe that was assayed in Fig 3 and will thus expose the other single strand of that DNA. Next, the newly formed ssDNA will be available to bind RNA and give rise to an artificial observation of strand invasion in vitro. This is a serious concern given three independent observations that were made by the authors: (1) In Extended Data Fig. 5c, the DNA is hardly seen in the presence of high PRC2 concentration points, which is consistent with nuclease activity. (2) The observation of strand invasion is dependent on Mg²⁺ (Extended Data Fig. 6c), which is also required for the activity of many nucleases. (3) The authors observed very little strand invasion activity of PRC2 toward a circular plasmid (Extended Data Fig. 5g), which is not as susceptible to exonuclease activity as the linearized DNA that was used in most of the assays. Since strand invasion is a major point in this paper, the authors should do the minimum in order to exclude DNA exonuclease activity in their protein preps. Specifically, the authors should incubate all the PRC2 complexes that were used in this study (fly PRC2 and human PRC2-EZH1 and PRC2-EZH2) with low nanomolar concentration of either fluorescently-labeled or radiolabeled short dsDNA probe of

approximately 100-200 bp, where exonuclease activity can be detected by electrophoresis (e.g. on acrylamide gels). A positive control should be included, e.g. an exonuclease from a commercial source, to demonstrate that the assay can detect exonuclease activity if present. Nuclease assays should be carried under the same experimental conditions (protein concentration, buffer, temperature, time etc.) as used in assays within Fig 3d-e, albeit without RNA. For complementation, the authors should repeat experiments in Fig 3d-e, where PRC2 will be replaced by a commercial DNA exonuclease that will yield an ssDNA and dNTPs as products. If strand invasion is really mediated by PRC2, the results should ideally show that PRC2 prep is devoid of nuclease activity and, independently, that exonuclease activity cannot lead to the same observation as in Fig 3d-e.

Response: Thank you for pointing out the possibility that strand invasion activity may be the result of a nuclease contaminant. As suggested, we have incubated *Drosophila* PRC2 and commercial endo- and exo-nucleases with oligonucleotides using strand invasion experimental conditions and were not able to detect degradation when incubated with PRC2 (Extended Data Fig. 9). We have also looked for ssDNA using EM and were not able to detect any long stretches of ssDNA when DNA was incubated with PRC2 (Fig. 4a-c). Finally, we have pre-incubated linear DNA with PRC2, before adding RNA and were not able to detect RNA-DNA hybrid formation unless PRC2 is added with the RNA (Fig. 4d-f). Based on these three independent experiments we conclude that contaminating nucleases in preparations of *Drosophila* PRC2 cannot explain the activity we observe.

We also include new data (Fig. 5, Extended Data Fig. 10) testing the ability of PRC2 to use DNA substrates with different ends (5' or 3' overhang, blunt end). This includes an analysis of templates prepared by digestion with EcoRI (Fig. 5). The RNAs used in these assays end 4 or 8 bases from the EcoRI digested end, depending on the orientation. RNA-DNA strand exchange is only observed with the RNA that initiates 4 bases from the end, but not with RNA initiating 8 bases from the end. This specificity further argues against a nuclease contaminant mediating the observed activity.

When we performed the same control experiments with hPRC2-Ezh1, we did not clearly detect oligonucleotide degradation. Using EM, hPRC2-Ezh1 does not generate ssDNA filament coated with SSB protein like exonuclease III-treated DNA, but smaller structures that may represent short ssDNA regions (and are not observed with *Drosophila* PRC2) were observed. In the pre-incubation experiment, the presence of hPRC2-Ezh1 in the second incubation increases RNA-DNA hybrid formation, but some RNA-DNA hybrid formation is observed with pre-treatment alone. These ambiguous results, along with the much higher concentrations of hPRC2-Ezh1 required to observe a much weaker activity than *Drosophila* PRC2, and the different reaction conditions for hPRC2-Ezh1 (37°C versus 30°C), which are more favorable for nuclease activity, mean that we cannot be completely sure that hPRC2-Ezh1 RNA-DNA hybrid formation is not the result of nuclease contaminants. We have therefore removed human PRC2 from the manuscript. We have included two summary figures of these experiments for the reviewers (Reviewer Figure 1, 2).

2. Ideally, strand invasion should be demonstrated using a non-linearized template: In Fig 3, the author used a linearized plasmid to demonstrate efficient strand invasion, but it is concerning that strand invasion experiments using non-linear plasmids (Extended Data Fig. 5g) showed poor strand invasion activity. The authors should quantify the variations between strand invasion to linearize vs circular

plasmids and comment on the potential reason(s) for these variations. If the authors have a concern regarding plasmid supercoiling as a potential negative factor for strand invasion, they can also compare non-linearized plasmid to non-linearized bacmid. If authors are concerned regarding another potential biophysical/biochemical restraint then they should test them experimentally. Leaving this observation without interpretation and testing is inappropriate, as one could expect strand invasion to take place also in the context of a circular dsDNA contract.

Response We have tried to induce RNA-DNA strand exchange by incubating PRC2 and RNA with negatively supercoiled, relaxed or nicked circular DNA, but have not succeeded. We do not know if PRC2 activity can only work on dsDNA with an end, or if it can cooperate with factors that open the DNA to induce RNA-DNA hybrids on closed DNA. Most published examples of protein-induced RNA-DNA hybrid formation have used linear substrates, or plasmids with a mismatched or ssDNA segment. One exception is TRF2, which can induce RNA-DNA hybrids using a supercoiled plasmid and RNA oligonucleotides. However, this plasmid is a specialized, repetitive telomere sequence that shows spontaneous hybridization of the RNA oligos (i.e. without addition of TRF2). In this case, the mechanism of strand invasion is believed to be an indirect effect of TRF2 on plasmid topology (Amiard et al., NSMB, 2007, Ref. 57). The other example is the viral protein ICP8. We have added a summary of the published observations to the Discussion to put the PRC2 activity in context. We also realized that the term “strand invasion” may not be the most appropriate for the reaction we observe, and have therefore chosen to switch to “RNA-DNA strand exchange” as a more precise description.

3. Binding assays for PRC1, as done, cannot support for variations in affinity: The authors state that R-loops are recognized by the PcG complex PRC1 in vitro. Yet, the binding “curve” for PRC1 in Fig 2e is actually linear, not a sigmoidal binding curve (an example for a good-looking sigmoidal binding curve is in Fig2f, where the same assay carried out for PRC2). A linear binding dependency, between protein concentration to fraction-bound nucleic acid, represents a titration experiment, which typically occurs where the probe (DNA) concentration is close to Kd concentration. In a titration experiment, the slope of the line in the plot does not represent affinity, but rather binding stoichiometry and/or the fraction of protein that is active in nucleic acid binding, e.g. see Ryder et al. 2008 (PMID: 18982286). Based on the information provided by the authors, DNA concentration seems to be around 2 nM, which is very close to the PRC1 concentration that was assayed: 5-20 nM. The problem likely occurs also in Extended Data Fig 5d. In order to assess variations in affinity, even qualitatively, the authors need to repeat the experiment in Fig 2e under conditions that will provide a sigmoidal binding curve, not a linear titration. This can be done by significantly reducing the probe concentration below Kd or (less optimal) to increase the salt concentration in order to increase Kd high above the probe concentration.

Response: The reviewer correctly points out that the conditions used for the binding assay are well outside what would be needed to measure correct affinities. We are unable to carry out this assay under the requisite conditions because the affinity of PRC1 for dsDNA is extremely high (0.2nM for a 150 bp oligo, and much higher for the large plasmid templates used here—a rough estimate is ~100x); using the template at the low concentrations needed to measure affinity would make it undetectable. PRC1 is also a difficult complex to prepare so that obtaining a high enough concentration to produce a full binding curve in this experiment is also not possible. We therefore decided to switch to short oligonucleotide substrates, where the affinities, although still high, allow radiolabelled probes to be

used in an appropriate range. Qualitative results from EMSA, and quantitative measurements from filter binding with the same probes indicate that both PRC2 and PRC1 bind R-loops formed in vitro more tightly than dsDNA. We may still overestimate the K_d for the R-loop and bubble DNA substrates (for PRC1) because of limitations on the concentration of probe needed to detect the signals well, but the difference is clear. Using synthetic substrates also allowed us to measure binding to an open DNA bubble, which is also higher than that for dsDNA. This adds insight into how PRC1 and PRC2 may recognize R-loops. These results are in Fig. 2 and Extended Data 5. We have removed the previous Extended Data Fig. 5 (EMSA of PRC1 and PRC2 binding to DNA and RNA).

4. The models presented in the model figures are not supported by the data: The authors do not provide any direct evidence to link the strand invasion activity that was observed in vitro to the function of PRC2 in cells. Given the complexity of the system and what is known so far, it is completely understandable if the authors would prefer to leave a comprehensive analysis in vivo to the scope of future studies. Yet, they should at least remove the model in Extended Data Fig. 10, which is completely imaginary, might be either right or wrong and is certainly not supported by the data, which did not bring into consideration the act of transcription, nucleosomes, epigenetic marks or anything else from this biological system in cells (with the exception of a basic association and correlative analyses in Fig 1). Furthermore, the authors should simplify the model presented in Extended Data Fig. 9. Specifically, although the three models proposed in Extended Data Fig 9a-c can all be tested biophysically in vitro, the authors didn't do it. Assuming the authors will be able to address concerns brought up above, the only model that is supported by the data is that PRC2 facilitates the formation of strand invasion. This is a significant finding, but lack mechanistic details and can, therefore, be represented in a simple one-step model.

Response: As requested, we have simplified our model, and summarized our in vitro observations and how we speculate they could be involved in PcG function in vivo. This model is presented in Fig. 7. We have removed the mechanistic models.

MINOR POINTS

5. The last sentence in the abstract state "... our findings suggest formation and recognition of non-canonical nucleic acid structures as an epigenetic mechanism". As exciting this idea is, it is a long stretch beyond the data. Supporting this statement will require the authors to perform experiments in cells to demonstrate that the inheritance of the repressed state through mitosis is dependent on the formation of PRC2-dependent R-loops and R-loop-dependent recruitment of PRC1. The authors can decide if they wish to perform these experiments or to revise that statement, but it is strongly recommended that they will do at least one of these.

Response: We have removed the statement.

6. Extended Data Fig. 1c: the table state "101460/17149 (61%)", but the math seems wrong. Is there a typo in "101460"?

Response: Thank you for pointing this out, we have corrected this table. Note that we have revised the numbers in this table as we realized that using the dm3 version of the *Drosophila* genome is not appropriate for transcript annotations, which have changed considerably in the dm6 version. We have therefore updated the data on R-loop overlap with genes and transcripts (Extended Data Fig. 1c, d and 2c) to the dm6 assembly.

7. Based on experiments in Fig 4e-f, the authors stated that RNA inhibits R-loop formation, but the data argue otherwise. Specifically, if instead of presenting (RNA-DNA)/(total RNA), in Fig 4f, the authors will present (RNA-DNA)/(total DNA) this will be evident.

Response: We have replotted all of our data as (RNA-DNA)/(total DNA) as suggested. This does indeed show that activity increases with increasing RNA and decreases with increasing DNA (Fig. 6). We have revised the results and conclusions accordingly.

8. The corresponding author has a solid and unquestionable experience from previous studies in the purification of PRC1. Yet, the purification of recombinant PRC1 in this study was modified and done under harsh conditions, with buffer containing 2 M KCl and 1 M Urea used for extraction from insect cells (Sf9) nuclei. The authors should provide evidence for complex assembly and solubility (ideally a gel filtration chromatography trace) and some evidence for enzymatic activity.

Response: The harsh conditions used in the PRC1 purification are used in the washing step, not to extract the complex. The complex is extracted from a standard (Dignam) nuclear extract (300mM KCl). The conditions are identical to what was published in 2001 (Francis et al.), except that we have added a nuclear purification step (through a sucrose cushion) prior to the extraction. The 2M KCl and 1M urea are used to wash the complex during the affinity purification (the 2M KCl wash is part of the original protocol). PRC1 purified using 2M KCl and 1M urea shows similar binding activity to PRC1 purified without the urea wash (We previously reported that PRC1 prepared without Ph, without the urea wash, is 20-30% active by filter binding (Lo et al., Mol. Cell, 2012). The two preparations used in Fig. 2 were measured as 31 and 40% active). PRC1 purified using this protocol is also highly active as a ubiquitin ligase for histone H2A. We have included an example of this activity for the reviewers (Reviewer Fig. 3), but will present this activity (which is not relevant to the current work) elsewhere. PRC1 is difficult to recover from size exclusion columns so we have not done this experiment, but the complex is well behaved in glycerol gradients. We have added protein gels of glycerol gradient fractionation of both PRC1 and PRC2 to Extended Data Fig. 4e, f. These show that the peaks of PRC1 without Ph (261kDa) and PRC2 (284kDa) migrate similarly, and we do not find evidence for a large amount of aggregated protein in the pellet (fraction 12 in Extended Data Fig. 4c, d).

9. In Fig 2C, the total amount of RNA seen on the gel is reduced as PRC1 concentration increased. Can the authors exclude RNase contamination that is associated with their PRC1 protein prep? Or is there any other explanation for that?

Response: We have removed these data from the manuscript, and replaced them with binding assays with both PRC1 and PRC2 using oligonucleotide substrates, as discussed above (point 3). R-loops associate with both the negatively supercoiled and relaxed form of the plasmid, but after incubation with PRC1 we don't observe the negatively supercoiled form of the plasmid; this is accompanied by the loss of R-loops. This reflects a strong effect of PRC1 on plasmid topology, an activity that is still under active investigation in our group. As the reviewer points out, we are aware that this further complicates the interpretation of the gradient binding data. These issues are eliminated by using the short (linear) oligonucleotide substrates to measure binding. In Extended Data Figure 5c and d, we have checked for nuclease degradation of the substrates used. We find a small amount of RNase activity (for PRC1) (on a free RNA substrate, which we did not use for binding assays), but no degradation of the R-loop template.

Reviewer #2 (Remarks to the Author):

In this manuscript, Alecki and colleagues investigate the possibility that R-loops might participate in the repressive functions of PcG proteins in Drosophila. They propose (i) that R-loops form at a subset of PREs sequences in vivo, (ii) that PRC1 binds to R-loops in vitro and (iii) that PRC2 is able to create R-loops in trans using a possible strand invasion activity. They investigate whether this strand invasion activity is conserved in human PRC2 proteins. The biological significance of this putative strand invasion activity was not investigated. Together, their data also seem to suggest that R-loops could form at PREs both co-transcriptionally (Figure 2) and in trans through the strand-invasion activity of PRC2 (Figure 3&4). The possible interplay between these putative -cis and -trans R-loops was neither investigated nor discussed. To synthesize their observations, the authors proposed that PRC2 creates R-loops in trans at PREs that are then recognized by PRC1. Whether the putative strand invasion activity of PRC2 is restricted to PREs sequences was neither explored nor discussed. The manuscript is relatively hard to follow and the starting hypothesis was not clearly explained. The figures would benefit from better labelling (especially when it comes to the hugely variable DNA and RNA concentrations used in the different assays). More importantly, the data lack important experimental controls. Overall, the authors failed to make a compelling case for their model.

Major points:

1. It is surprising that so many R-loops would form over "repressed" transcription units. At PREs, whether these R-loops are co-transcriptional or formed in "trans" by the putative strand invasion activity of PRC2, these loci must be transcribed at some point. A comparison of DRIP-seq maps to RNAP2 ChIP-seq maps would therefore have been more appropriate than a comparison to RNA-seq data as presented here, because RNA-seq only measures steady-state levels of soluble RNAs. Published RNAP2 ChIP-seq data might be available.

Response: Our analysis of DRIP-seq data was carried out with the dm3 version of the *Drosophila* genome, which is appropriate for comparison with ChIP-seq data. However, we realized that annotated transcripts are much more extensive in the new dm6 genome, particularly for non-coding transcripts. We have therefore repeated the overlaps between DRIP-seq data and annotated transcripts using the dm6 genome (Extended Data Fig. 1c, 2cd). This shows a higher overlap between DRIP-seq peaks and

annotated transcripts, although in all data sets, there are DRIP-seq peaks that either do not overlap annotated transcripts, or only have annotated transcripts in the opposite orientation. We also analyzed overlap between PREs and RNA PolIII ChIP-seq data, as requested, and find that PREs that form R-loops are more likely to have PolIII signal (Extended Data Fig. 2d). In the discussion, we mention the possibility that some R-loops may be formed in *trans*. Because we do not have any functional data to address this interesting possibility in vivo, we have kept our discussion of it minimal.

2. The demonstration that co-transcriptional R-loops form at PREs in vitro is not compelling.

a. One wonders how the authors can claim that their RNA-DNA hybrids are resistant to RNase A treatment when their data clearly show that their hybrid signal is sensitive to increasing RNase A concentrations (Figure 2b, Figure 3hi). Did the authors use sufficiently high salt concentrations in their reactions to limit the RNase H activity of RNase A?

Response: We have used 300 mM NaCl for RNase A treatment of RNA-DNA hybrids to limit the RNase H activity of RNase A. We cannot exclude the possibility that some of the R-loops have been degraded by RNase A. Because we do not know the size of the R-loops that are formed, it is possible that the reduction in signal reflects digestion of parts of the RNA that are not part of the R-loop. As noted previously, the RNaseH activity of RNase A makes it an imprecise tool for quantitative analysis of ribonuclease sensitivity (Sanz & Chedin, Nature Protocols 2019). We also observe S9.6 reactivity on dot blots of the transcribed templates. However, all of the data with in vitro transcribed templates have been removed from the manuscript. We have included an immunoprecipitation experiment (Extended data Fig. 6c, d) demonstrating that the RNA-DNA hybrids induced by PRC2 are recognized by the S9.6 antibody.

b. The radiolabelled RNA seems to associate with both the supercoiled and the open form of the plasmid (Figure 2b, SC and R/N forms should be labelled on the figure). This makes little sense, as hybrid formation is expected to relax the plasmid. The authors justify this by saying that R-loops are probably small (“a few hundred bp”, legend of the Extended data Fig. 4) but “a few hundred bp” seem already to be quite big for R-loops (R-loops of around 200 bp were largely sufficient to completely relax pFC53 according to a recent publication by the Chédin lab). This might suggest that R-loops formed at PREs are indeed very small. It would therefore be nice to repeat some of their experiments with well-described R-loops of reasonable sizes, such as those formed in pFC53 (see below).

Response: The reviewer has raised a good point that we do not know the size of the R-loops, or where they are formed in the in vitro transcription assays. In the revised manuscript, we do not use the in vitro transcribed templates for any experiments, so have removed the figures describing their characterization. As the reviewer points out, assessing the R-loop forming potential of PRE sequences in vitro will require a detailed and precise analysis, which is beyond the scope of the current work.

c. Their vg-PRE insert has been cloned 200 bp downstream of the T7 promoter according to the legend of Extend data Figure 6. The authors should make sure that this 200 bp sequence cannot contribute to the

weak R-loop formation that they detected in their in vitro transcription assay, for example by using a T7-containing empty vector as negative control.

Response: see below (point d).

d. The authors should use dot blots with the S9.6 antibody to independently and more precisely quantify R-loop formation at *vg*-PRE and *Airn*.

Response: We have looked at the R-loop forming potential of *vg*, the empty vector (pETblue1) but also pFC53 and pFC53 without the R-loop forming region using the two transcription protocols. On gels, we observed that pFC53 shows a stronger change in plasmid topology when transcribed compared to *vg*-pETblue1 and pETblue1 empty vector using either transcription protocol. Plasmid relaxation was reversed by treating the sample with RNase H but not with RNase A for all plasmids. We confirmed the presence of R-loops by dot blot using S9.6 antibody. We detected similar levels of R-loops on all plasmids tested and these R-loops were sensitive to RNase H but not RNase A. Given that all experiments using in vitro transcribed plasmids have been removed from the manuscript, we have not included these data.

3. In Figure 2ef, what does “DNA” mean? Has the “DNA” been transcribed? Or does it correspond to the mix of plasmids with and without hybrids after transcription? In the binding assays (Figure 2cd), a non-transcribed plasmid and/or a RNase H-treated plasmid must be used as specificity controls (this should really be the “DNA” in figure 2ef).

Response: The DNA corresponds to the mix of plasmids with and without hybrids after transcription. Because we expect R-loop formation under the conditions used for these assays to be quite low, we have quantified the total DNA (with and without hybrids) so that we can quantify both RNA-DNA (radiolabelled) and DNA from the same reactions. The reviewer raises an important point that we do not know what fraction of the total DNA has formed R-loops in these reactions. We have replaced the gradient binding assay for both PRC1 and PRC2 with EMSA and filter binding with oligonucleotide substrates, which gives more precise results (and indicates that PRC2 can recognize R-loops). These data are in the new Fig. 2 and Extended Data Fig. 5. Please see the response to Reviewer 1's 3rd point for a full explanation.

*It looks as if PRC1 associates preferentially with the relaxed form of the plasmid: is it because R-loops are longer on the relaxed form of the plasmid (the plasmid has been relaxed by the long R-loops), or is it simply that PRC1, contrary to PRC2, has a greater affinity for relaxed than for supercoiled DNA? If the latter were true, the apparent preferential binding of PRC1 to “hybrids” would only be an indirect consequence of R-loop driven plasmid relaxation. This could easily be assessed with nicking enzymes. However, a better experiment would be to cut the plasmid in half after transcription (topology would not be an issue anymore): if the authors could show that PRC1 has indeed a stronger affinity for the R-loop containing fragment than for the plasmid backbone, the demonstration that PRC1 binds to R-loops would be a lot more compelling (this could be quantified by qPCR). As it stands, the demonstration that PRC1 binds to R-loops is not convincing. To strengthen their case even further, the authors should also assess the binding of PRC1 and PRC2 to well-described R-loops such as those formed at *Airn*.*

Response: We agree that the binding assays with PRC1 had multiple issues, as also pointed out by Reviewer 1. We have replaced the sucrose gradient binding assay with EMSA and filter binding, as described immediately above (Fig. 2 and Extended Data Fig. 5).

4. The experiment showing that PRC2 promotes R-loop formation (Extended Data Figure 5g) should be quantified and should be confirmed using the S9.6 antibody (IP followed by qPCR). Could PRC2 really do this without the need for ATP?

Response: We have removed Extended Data Figure 5. We used the S9.6 antibody to confirm that PRC2 induces the formation of RNA-DNA hybrids (on linear DNA templates) by immunoprecipitating the reaction products with the S9.6 antibody (Extended Data Fig. 6c, d). To confirm that ATP is not required for RNA strand exchange, we incubated PRC2 or the RNA with apyrase and we have used them in RNA exchange assays. RNA strand exchange was observed at similar levels with both apyrase treated or not treated RNA and protein (Extended Data Fig. 7e-g).

5. The different “strand invasion” reactions (Figures 3&4 and Extended data Figure6&7) are confusing because

(i) the concentrations of DNA and RNA vary widely from one experiment to the next;

Response: We intentionally titrated the RNA and DNA substrates of the reaction. However, we realized that these data were not plotted in an informative way, and in fact, as pointed out by Reviewer 1 (point 7), the data presentation actually obscured the result. We have changed all of our graphs to show (RNA-DNA)/DNA (Fig. 6). The other experiments were carried out under our standard reaction conditions, which are 0.31 nM DNA and 0.18 nM RNA.

(ii) it is surprising that PRC2 would be able to promote strand invasion without energy (no ATP was added to the reactions?)

Response: As described in our response to point 4, we have used apyrase treatment to rule out contaminating ATP in the RNA-DNA hybrid formation assay (Extended data Fig. 7e-g). We also note that for other proteins that are able to induce RNA-DNA hybrid formation in vitro (Rad52/RecA, PALB2, TRF2, CE, ICP8), ATP is not required. We now detail the comparison of PRC2 with these proteins in the Discussion.

(iii) the efficiency of the strand invasion reaction appears to be very weak and even weaker in human proteins. Even when the authors used only minute amount of RNA (fmol, Figure 4), a very large excess of PRC2 was not able to “recombine” the totality of that RNA (max 60%), even when the DNA concentration was increased to provide more templates to “recombine” with. One therefore wonders at the biological significance of these observations.

Response: We have removed hPRC2 from the manuscript, as described in point 1 in our response to Reviewer 1. We agree that the ratio of PRC2 to substrates used in our assay is high. At this time, we do not know how this reaction works, so we do not know why this is the case. Part of the explanation may be that the substrates we are using are quite large (~2kb for the RNA and ~5kb for the DNA) so that each substrate molecule is likely to bind multiple copies of PRC2. Whether this is part of the reaction

mechanism (for example through formation of a protein-DNA or protein-RNA filament similar to Rad52 (Mazin et al., 2017, McDevitt et al., 2018)) is entirely unknown at this time. We have stated nucleic acid concentrations as molar amounts of molecules. However, some reports in the literature instead use the concentration of base pairs/nucleotides for DNA and RNA respectively when reporting similar activities. For example, Kashahara et al. (2000) report using 4 μM RecA with 15 μM (bp) dsDNA (circular DNA with an unpaired region) and 0.8 μM (ntd) RNA to induce R-loop formation in vitro; similarly, Zaitsev & Kowalczykowski (2000) used a ratio of 1 RecA to 10 bp. If we convert our reaction conditions to this metric, we use 25-400 nM PRC2 with 0.31 nM DNA *5000 bp=1.55 μM DNA bp and 0.18 nM *1600 ntd =288 nM RNA ntd.

In addition, Figures 3g/4d/S6d indicate that above ~30 nM of PRC2, the reaction seems to reach a plateau. Yet, the authors have made most of their measurements (Figures 4) at concentrations of PRC2 higher than 30 nM and they might have missed important observations as a result.

Response: In Figure 3g 30 nM PRC2 is approximately the mid point of the reaction, and Fig. 4b (now Extended Data Fig. 7b) indicates that 50 nM PRC2 is still in the dynamic range with respect to protein concentration. Although we agree that the lower protein concentrations should reveal the most significant changes in the reaction behavior, we think that the titrations provided are sufficient for a basic characterization of the effect of titrating RNA and DNA.

The specificity controls that the authors used (Sxc and NFY) have a much lower affinity for RNA than PRC2 according to their own measurements. It would be more appropriate to use as specificity control a protein that has a comparable affinity for RNA, such as PRC1, which the authors have produced.

Response: We have included these control proteins simply to show that not all RNA and DNA binding proteins purified from Sf9 cells in our lab have RNA-DNA hybrid forming activity. Sxc and NFY were used at concentration were both show a complete binding of RNA and DNA by EMSA (Extended Data Fig. 8a-e).

Finally, how do the authors explain that the RNA that was not “recombined” with the DNA template did not associate with the massive excess of PRC2 considering the great affinity of PRC2 for RNA (no shift of the remaining RNA to the top of the gel like in their other EMSA assay (Extended data Figure 5b))?

Response: As noted above, Extended data Figure 5b is no longer part of the manuscript. However, the reason we do not observe RNA binding in this experiment is that it is not an EMSA. Instead, after the PRC2 incubation, the reactions were extensively digested with proteinase K so that we could observe the migration of the plasmid, plasmid-associated RNA, and free RNA. We apologize that this figure was not well explained in the original submission.

Minor comments:

1. A better explanation of the starting hypothesis would be helpful.

Response: We have substantially revised the manuscript.

2. The Discussion is very limited and should better mention the work recently published by Konstantina Skourti-Stathaki and Ana Pombo in Molecular Cell.

Response: We have substantially revised the Discussion, and cited the Skourti-Stathaki paper multiple times.

3. The idea that non-canonical nucleic acid structures, and especially R-loops, could convey epigenetic regulation has already been suggested by the work of Andrés Aguilera.

Response: We have modified our abstract to remove the claim of novelty for this suggestion, as we agree that this overlooks substantial work on chromatin and R-loops, including the pioneering work of Aguilera connecting histones and histone phosphorylation to R-loops. While we were not able to discuss previously described links between R-loops and epigenetic regulation comprehensively, we have cited recent reviews (including from Aguilera) which cover this topic.

a

b

c

Reviewer Figure 1 hPRC2 Tests for nuclease

Reviewer Figure 1 Testing hPRC2-Ezh1 preparation for nuclease activity. a. Scheme of the experiment testing whether PRC2 preparations contain contaminant nuclease activity. b-f. b. Representative gels of phosphorylated dsDNA or ssDNA oligonucleotides after incubation with hPRC2-Ezh1, T7 endonuclease I, exonuclease I, λ exonuclease, or exonuclease III (n=3) incubated under strand exchange reaction conditions. Final panel shows dPRC2 incubated with oligonucleotides under the same conditions as for hPRC2-Ezh1. c. Representative gels of DNA and Cy5-labelled RNA after incubation with the indicated nucleases under strand exchange reaction conditions (n=2). In all panels the titrations are [PRC2]: 25-400 nM, [T7 endonuclease I]: 0.00004-4 units, [exonuclease I]: 0.00008-8 units, [λ exonuclease]: 0.00002-2 units, [exonuclease III]: 0.0004-40 units.

Reviewer Figure 2 Control experiments performed with hPRC2-Ezh1 a, b. Pre-incubation of hPRC2-Ezh1 templates with DNA allows a small amount of RNA-DNA hybrid formation without addition of hPRC2-Ezh1 during the second incubation. c. A small number of SSB coated templates are visible in preparations of hPRC2-Ezh1 treated DNA (top panel) (arrowheads show examples; arrows show examples of DNA molecules without SSB coating), although clearly less than observed with DNA treated with ExoIII under the same conditions (bottom panel).

Reviewer Figure 3 PRC1 Δ Ph is an active E3 ligase for histone H2A

Reviewer Figure 3 Ubiquitin ligase activity of PRC1 Δ Ph. a,b Cy3 (H2A) scan and SYPRO Ruby stain for histone ubiquitylation reaction with PRC1DPh and a chromatinized plasmid showing that E3 activity requires E1, E2, Ub, ATP, and PRC1 Δ Ph. Asterisks indicate spurious contaminants in one tube. c. Titration of PRC1 Δ Ph E3 ligase activity on a chromatin template (80nM nucleosomes) with Cy3 labelled histone H2A.

REVIEWERS' COMMENTS:

Reviewer #1 (Remarks to the Author):

The authors addressed critical issues in their work, moderated some of the statements and removed some of the less convincing data, including experiments with human PRC2 and plasmids. The finding that both fly PRC1 and PRC2 bind to R-loops and facilitate the formation of DNA-RNA hybrids in vitro is significant. In this sense, this work complementing (not redundant with) recent literature and worthy a publication. There are several minor points that should better be addressed before publishing this:

1. With the exclusion of experiments using human PRC2, the scope of the work has now been reduced to fly PRCs and PREs. With fly and PREs being a key model system for the study of polycomb biology, the findings are still significant but the right thing would be to state this information in the title. This is important as the mammalian PcG system is more complex, in terms of protein subunits, and does not appear to be heavily relied on PREs, with only a few exceptions. Hence, stating fly/*Drosophila* in the title will leave an open path for future works by the authors (and others) to explore the findings in a mammalian context and will protect the authors in case that another mechanism will be identified in mammals.

2. The authors state in the abstract that "The PcG complexes PRC1 and PRC2 can recognize R-loops in vitro". But Fig 2 indicates that PRCs actually bind to both R-loops and DNA bubble, with the latter being sufficient for high-affinity interactions. Hence, the RNA in the R-loop is dispensable for high-affinity interactions. Since this is a key point in this work, it would be most appropriate to mention in the abstract that the DNA bubble is sufficient for high-affinity interactions with PRC2; currently, only R-loops are mentioned, but PRC-R-loop interactions could simply be the consequence of interactions between PRCs to the DNA bubble.

3. In Fig 2c, when quantifying the affinities of PRCs to the R-loop and the DNA bubble, the authors reported Kds of <0.3 nM and <0.03 for PRC2 and PRC1, respectively. As the probe concentration is 0.01 nM (i.e. very close to Kds measured for PRC1), Kds could actually be lower than what quantified, and therefore the data cannot determine if the R-loop or the DNA-bubble is the preferred ligand for PRC1. This should minimally be mentioned in the discussion.

Since these are all minor points that can be addressed textually, I don't think that another round of review is needed before publication.

Reviewer #2 (Remarks to the Author):

The authors have submitted a significantly revised manuscript. One could almost say in fact that this is a new submission altogether, because a significant number of experiments have been removed and replaced by others. The authors have improved their demonstration that *Drosophila* PRC2 might provide

a limited RNA-DNA strand exchange activity in vitro but they were unable to establish conclusively that this activity is also conserved in the human PRC2 complex. It is also still unclear how long the “newly-formed” RNA-DNA hybrids are, although the authors now show that they are long enough to be recognized by the S9.6 antibody. More importantly, the authors still failed to provide evidence that this activity is relevant in vivo. It is a concern that this activity requires a free DNA end in vitro as free DNA ends are likely to be sparse at PREs in vivo. It seems therefore unlikely that PRC2 will contribute significantly to the formation of RNA-DNA hybrids in vivo at PREs, contrary to what the authors suggest in their Discussion. I believe that this detailed in vitro characterisation of PRC2 activity would be more adapted to a more specialized journal, such as JCB or NAR.

REVIEWERS' COMMENTS:

Reviewer #1 (Remarks to the Author):

The authors addressed critical issues in their work, moderated some of the statements and removed some of the less convincing data, including experiments with human PRC2 and plasmids. The finding that both fly PRC1 and PRC2 bind to R-loops and facilitate the formation of DNA-RNA hybrids in vitro is significant. In this sense, this work complementing (not redundant with) recent literature and worthy a publication. There are several minor points that should better be addressed before publishing this:

1. With the exclusion of experiments using human PRC2, the scope of the work has now been reduced to fly PRCs and PREs. With fly and PREs being a key model system for the study of polycomb biology, the findings are still significant but the right thing would be to state this information in the title. This is important as the mammalian PcG system is more complex, in term of protein subunits, and does not known to be heavily relied on PREs, with only a few exceptions. Hence, stating fly/Drosophila in the title will leave an open path for future works by the authors (and others) to explore the findings in a mammalian context and will protect the authors in case that another mechanism will be identified in mammals.

Response: We have specified that it is Drosophila PRC2 in the title.

2. The authors state in the abstract that “The PcG complexes PRC1 and PRC2 can recognize R-loops in vitro”. But Fig 2 indicates that PRCs actually bind to both R-loops and DNA bubble, with the latter being sufficient for high-affinity interactions. Hence, the RNA in the R-loop is dispensable for high-affinity interactions. Since this is a key point in this work, it would be most appropriate to mention in the abstract that the DNA bubble is sufficient for high-affinity interactions with PRC2; currently, only R-loops are mentioned, but PRC-R-loop interactions could simply be the consequence of interactions between PRCs to the DNA bubble.

Response: We indicate that PRC2 and PRC1 can recognize both R-loop and open DNA bubble in the abstract. Finally, we state that we don't know which part of the R-loop or open DNA is recognize by PcG complexes (ssDNA, structured DNA, ssDNA/dsDNA junction...) and that it is unlikely that PcG complexes recognize RNA-DNA hybrids.

3. In Fig 2c, when quantifying the affinities of PRCs to the R-loop and the DNA bubble, the authors reported Kds of <0.3 nM and <0.03 for PRC2 and PRC1, respectively. As the probe concentration is 0.01 nM (i.e. very close to Kds measured for PRC1), Kds could actually be lower than what quantified, and therefore the data cannot determine if the R-loop or the DNA-bubble is the preferred ligand for PRC1. This should minimally be mentioned in the discussion.

Response: We have modified the Results to point out the limitations to the binding measurements.

Since these are all minor points that can be addressed textually, I don't think that another round of review is needed before publication.

Reviewer #2 (Remarks to the Author):

The authors have submitted a significantly revised manuscript. One could almost say in fact that this is a new submission altogether, because a significant number of experiments have been removed and replaced by others. The authors have improved their demonstration that *Drosophila* PRC2 might provide a limited RNA-DNA strand exchange activity in vitro but they were unable to establish conclusively that this activity is also conserved in the human PRC2 complex. It is also still unclear how long the “newly-formed” RNA-DNA hybrids are, although the authors now show that they are long enough to be recognized by the S9.6 antibody. More importantly, the authors still failed to provide evidence that this activity is relevant in vivo. It is a concern that this activity requires a free DNA end in vitro as free DNA ends are likely to be sparse at PREs in vivo. It seems therefore unlikely that PRC2 will contribute significantly to the formation of RNA-DNA hybrids in vivo at PREs, contrary to what the authors suggest in their Discussion. I believe that this detailed in vitro characterisation of PRC2 activity would be more adapted to a more specialized journal, such as JCB or NAR.

Response: We have emphasized in our discussion that we don't know yet the extent to which PRC2 contribute to R-loop formation at PREs in vivo and that in vitro PRC2 cannot induced strand invasion on circular DNA suggesting we are missing another component (DNA breaks/active transcription/protein to open the DNA bubble).